# Towards Optimization-Friendly Binary Neural Network

**Nianhui Guo**                                                        *nianhui.guo@hpi.de*
*Hasso Plattner Institut, University of Potsdam, Germany*

**Joseph Bethge**                                                      *joseph.bethge@hpi.de*
*Hasso Plattner Institut, University of Potsdam, Germany*

**Hong Guo**                                                            *hong.guo@hpi.de*
*Hasso Plattner Institut, University of Potsdam, Germany*

**Christoph Meinel**                                                *christoph.meinel@hpi.de*
*Hasso Plattner Institut, University of Potsdam, Germany*

**Haojin Yang**                                                        *haojin.yang@hpi.de*
*Hasso Plattner Institut, University of Potsdam, Germany*

**Reviewed on OpenReview:** *https://openreview.net/forum?id=4Hq816XDDG*

## Abstract

Binary neural networks (BNNs) are a promising approach for compressing and accelerating deep learning models, especially in resource-constrained environments. However, the optimization gap between BNNs and their full-precision counterparts has long been an open problem limiting their performance. In this work, we propose a novel optimization pipeline to enhance the performance of BNNs. The main approach includes three key components: (1) BNext, a strong binary baseline based on an optimization-friendly basic block design, (2) knowledge complexity, a simple yet effective teacher-selection metric taking the capacity gap between teachers and binary students under consideration, (3) consecutive knowledge distillation (CKD), a novel multi-round optimization technique to transfer high-confidence knowledge from strong teachers to low-capacity BNNs. We empirically validate the superiority of the method on several vision classification tasks CIFAR-10/100 & ImageNet. For instance, the BNext family outperforms previous BNNs under different capacity levels and contributes the first binary neural network to reach the state-of-the-art 80.57% Top-1 accuracy on ImageNet with 0.82 GOPS, which verifies the potential of BNNs and already contributes a strong baseline for future research on high-accuracy BNNs. The code is publicly available at `https://github.com/hpi-xnor/BNext`.

## 1 Introduction

Deep neural networks (DNNs) have made remarkable progress in almost all areas of AI research in recent years (He et al., 2016; Simonyan & Zisserman, 2014; Szegedy et al., 2015; Huang et al., 2017; Tan & Le, 2019; Dosovitskiy et al., 2020; Wang et al., 2020; Liu et al., 2022). However, the development of modern neural network architectures is usually accompanied by an increase in computational budget, memory usage, and energy consumption, which can hinder their deployment on resource-constrained edge devices such as cell phones, mini-robots, AR glasses, and autopilot systems (Simonyan & Zisserman, 2014; Szegedy et al., 2015; He et al., 2016; Huang et al., 2017; Tan & Le, 2019; Dosovitskiy et al., 2020; Wang et al., 2020; Liu et al., 2022). Thus, there exists a significant technical gap between the evolutionary trend of modern neural networks and edge applications. To bridge this gap, it is necessary to co-design optimization strategies from both sides to maximize the potential of deep neural networks on the edge.

Motivated by this, various techniques have been developed to compress and accelerate modern neural networks, such as Network Pruning (Han et al., 2015), Knowledge Distillation (Hinton et al., 2015), Compact Architecture Design (Howard et al., 2017), and low-bit Quantization (Jacob et al., 2018; Banner et al., 2019). Among them, Binary Neural Networks (BNNs) have gained attention due to their potential on edge devices (Courbariaux et al., 2016; Rastegari et al., 2016; Liu et al., 2018; Martinez et al., 2020; Liu et al., 2020b; Zhang et al., 2021). BNNs quantize weights and activations to 1-bit values ($+1$ or $-1$), which can reduce memory requirements by $32\times$ and achieve a theoretical speedup of $58\times$ on the CPU by replacing the floating-point dot products with XNOR and bit-counting operations (Courbariaux et al., 2016; Rastegari et al., 2016). Despite the apparent efficiency advantages, binary neural networks (BNNs) have long faced optimization difficulties and accuracy degradation problems (Courbariaux et al., 2016; Rastegari et al., 2016; Liu et al., 2018; Martinez et al., 2020; Liu et al., 2020b; Zhang et al., 2021). While previous works have attempted to narrow the accuracy gap to full-precision models such as ResNet-18 and ResNet-50, the latest BNNs still lag behind their full-precision counterparts by approximately 10% (Dosovitskiy et al., 2020; Liu et al., 2022; 2021a). This is due to their significantly reduced representation capacity (by a factor of $3.4 \times 10^{38}$) (Tu et al., 2022) and increased optimization difficulty. To solve the non-differentiability problem in backpropagation, BNN training relies on gradient estimation techniques, such as the Straight-Through Estimator (Courbariaux et al., 2016). Moreover, BNNs have a much coarser loss landscape, making optimization more challenging than full-precision networks (see Fig. 1b). These issues demand specific architectural designs and optimization methodologies for highly accurate BNN results.

In this paper, we first propose BNext, a novel binary architecture that serves as the baseline for high-accuracy exploration. To address the limitations of previous BNNs and improve the representation capacity, we introduce a novel binary processing unit, called Info-RCP, which incorporates an adaptive information re-coupling structure. Different from previous methods (Rastegari et al., 2016; Martinez et al., 2020; Zhang et al., 2021) that simply use channel-wise scaling to mitigate information loss, we find that these approaches do not consider the semantic gaps before and after binary convolution, resulting in suboptimal information reconstruction. Our method scales and shifts the binary convolution outputs using an additional BatchNorm and PReLU layer first, followed by an attention branch that combines information before and after the binary unit. We also introduce a new shortcut design for Infor-RCP to enable element-wise attention (ELM-Attention) mechanism and enhance the block capacity, inspired by the regularized architecture design of recent vision transformers (Han et al., 2021; Liu et al., 2021a).

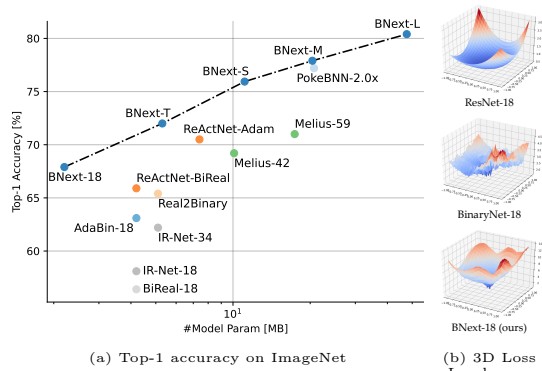

(a) Top-1 accuracy on ImageNet

(b) 3D Loss Landscape

Figure 1: (a) Comparison with SOTA binary neural networks on ILSVRC-2012 ImageNet regarding accuracy and model param. (b) 3D loss landscape visualization (Li et al., 2018) of full-precision ResNet-18 (He et al., 2016), BinaryNet-18 (Courbariaux et al., 2016) and our BNext-18.

It should be noted that the additional computational overhead introduced by the ELM-Attention mechanism is almost negligible. Finally, We propose an effective architecture scaling strategy and construct the BNext family by stacking the basic blocks based on it. We empirically evaluate the superiority of BNext design principle by visualizing the loss landscape (Fig. 1b). BNext holds a much smoother and more complete loss landscape compared to a traditional binary ResNet-18 and is pretty close to a full-precision ResNet-18.

From the optimization aspect, though Knowledge Distillation (KD) is already a commonly used technique in previous works (Martinez et al., 2020; Liu et al., 2020b; 2021b; Zhang et al., 2021), rarely work has paid attention to the impacts and principles of teacher selections for BNNs. During our empirical exploration, we show that not all teachers teach equally to binary students and thus present a specific metric, *Knowledge Complexity*, as a simple yet effective indicator for teacher model selection. we show that directly transferring Knowledge from a higher knowledge-complexity teacher is less efficient for binary students, which can suffer from counter-intuitive overfitting problems. Based on the observation and proposed metric, we build consecutive knowledge distillation (CKD), a novel multi-round optimization technique, where the high-complexity knowledge is assembled and regularized by low-complexity knowledge during the training. We evaluate the

proposed optimization pipeline on the challenging ImageNet classification task. As shown in Fig. 1a, the BNext family outperforms previous BNNs under different model size levels and contributes the first binary neural network baseline to reach the 80% accuracy level on ImageNet.

Our main contributions can be summarized as follows:

1. *BNext*, a novel optimization-friendly binary architecture that serves as a strong baseline for high-accuracy BNN exploration.

2. *Consecutive Knowledge Distillation*, an effective and practical technique to alleviate the knowledge gap during binary model optimization.

3. Evaluation on ImageNet contributes the first BNN with 80.57% Top-1 accuracy, which provides a meaning strong baseline for future high-accuracy BNNs research.

## 2 Related Work

**BNNs Optimization.** Binarization of neural networks reduces computational and memory requirements, but optimizing binary neural networks (BNNs) is challenging due to their non-differentiability and loss of information (Courbariaux et al., 2016). BinaryNet solved this challenge by using the straight-through-estimation (STE) technique (Courbariaux et al., 2016), and subsequent works have attempted to improve optimization using different variants of STE. BOP proposed a specialized optimizer to flip binary states (Helwegen et al., 2019), while Real2BinaryNet used knowledge distillation in a three-stage process (Martinez et al., 2020), and other works simplified it to a two-stage process (Liu et al., 2020b; Zhang et al., 2021). Additionally, Liu et al. explored how training strategies such as optimizers and weight decay aid BNN optimization (Liu et al., 2020b). In this paper, we analyze optimization-friendly module designs in BNNs to further improve optimization, helping us understand why BNNs are difficult to optimize and how to construct an optimization-friendly architecture.

**Knowledge Distillation.** In knowledge distillation (Hinton et al., 2015), dark knowledge is transferred from a pre-trained teacher model to a student model. Recent works have aimed to gain a deeper understanding of this technique. For instance, it has been discovered that there exists a large discrepancy between the predictive distributions of the teacher and the student (Stanton et al., 2021). Beyer (Beyer et al., 2022) highlight that certain implicit design decisions can significantly impact the efficacy of distillation. Park (Park et al., 2021) improve the distillation process by creating student-friendly teachers and revealing the presence of knowledge mismatch between the teacher and student. Moreover, modern binary neural network optimization (Martinez et al., 2020; Liu et al., 2020b; Zhang et al., 2021) heavily relies on knowledge distillation techniques (Hinton et al., 2015; Shen & Savvides, 2020). In particular, the soft label from a pre-trained teacher provides a more fine-grained supervision signal than a one-hot label. This encourages the research community to explore the BNN-specific optimization pipeline for improved model performance.

**Modern Neural Network Optimization.** Modern deep neural networks (Tan & Le, 2019; Dosovitskiy et al., 2020; Liu et al., 2021a; 2022) owe their success in image classification not only to architecture design ideas, such as attention mechanisms, but also to a collection of state-of-the-art optimization techniques. Data augmentation strategies, like MixUp (Zhang et al., 2017), CutMix (Yun et al., 2019), RandAugmentation (Cubuk et al., 2020), and Augmentation Repetition (Hoffer et al., 2020), have been shown to benefit the generalization of regular 32-bit networks (Ridnik et al., 2022). However, it remains an open question whether these strategies can be applied to highly accurate BNNs, or if BNN-specific augmentation techniques are necessary. This paper presents a fair and comprehensive empirical study of modern optimization techniques for BNNs in large-scale image classification tasks. The study demonstrates the importance of revisiting BNN-specific augmentation strategies and identifies potential solutions.

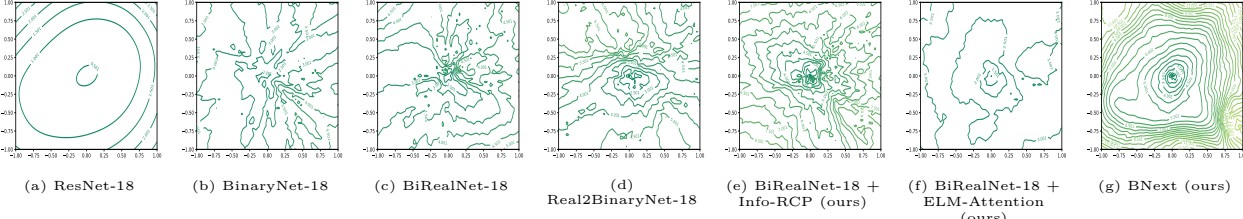

|     |     |     |     |     |     |     |
|-----|-----|-----|-----|-----|-----|-----|
| (a) ResNet-18 | (b) BinaryNet-18 | (c) BiRealNet-18 | (d) Real2BinaryNet-18 | (e) BiRealNet-18 + Info-RCP (ours) | (f) BiRealNet-18 + ELM-Attention (ours) | (g) BNext (ours) |

Figure 2: The loss landscape visualization of popular BNNs (contour line view) based on the CIFAR10 dataset. We follow the setting of (Li et al., 2018). "Info-RCP" and "ELM-Attention" are the module designs in this paper. The corresponding 3D views are available in the supplementary material.

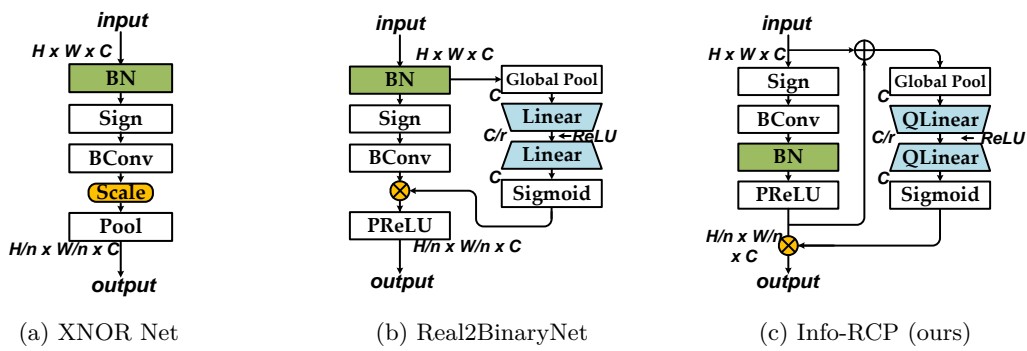

(a) XNOR Net     (b) Real2BinaryNet     (c) Info-RCP (ours)

Figure 3: Binary convolution module design comparison among XNOR-Net, Real2BinaryNet and Ours.

## 3 BNext Architecture Design

This section introduces the BNext architecture design, its motivation, and details. In Section 3.1, we visualize the loss landscape of popular binary neural networks, offering valuable insights. Additionally, we present the two core modules of BNext and an overview of the BNext model family's structure in Section 3.2.

### 3.1 Visualizing the Optimization Bottleneck

Binary neural networks are a type of lightweight neural network that consists of stacked 1-bit convolutions. During forward propagation, both the inputs $x_r^t$ and the proxy weights $w_r^t$ in each 1-bit convolution are binarized into $\pm 1$ using the sign function before the dot product computation. In backward propagation, the gradient of the hardtanh function approximates the sign function (Hubara et al., 2016), enabling us to compute the gradient of $w_b^t$ and optimize the weights $w_r^t$ using end-to-end SGD. The gradient of $w_b^t$ is accumulated in the proxy weights $w_r^t$ during each iteration (Courbariaux et al., 2016). This process can be mathematically formulated as follows:

$$x_b^t, w_b^t = \text{binarize}\left(x_r^t, w_r^t\right) = \left(\text{sign}\left(x_r^t\right), \text{sign}\left(w_r^t\right)\right), \tag{1}$$

$$w_r^{t+1} = w_r^t - \alpha \cdot \frac{\partial \ell^t}{\partial w_r^t}, \quad \frac{\partial \ell^t}{\partial w_r^t} = \frac{\partial \ell^t}{\partial w_b^t} \cdot \frac{\partial w_b^t}{\partial w_r^t}, \tag{2}$$

where $\frac{\partial w_b^t}{\partial w_r^t} = 1_{||w_r^t|| \leq c}$, represents the Straight-Through-Estimation (STE) strategy. $r$, $b$, and $t$ represent real-valued variables, binary variables, and iteration numbers, respectively.

Despite the Straight-Through-Estimation (STE) strategy partially addressing the non-differentiable problem, Binary Neural Networks (BNNs) suffer from severe accuracy degradation due to the binary representation and gradient mismatch problem (Courbariaux et al., 2016; Rastegari et al., 2016). To address these bottlenecks, several subsequent designs (Rastegari et al., 2016; Liu et al., 2018; Martinez et al., 2020; Bethge et al., 2020; Liu et al., 2020b) have been proposed to improve capacity and optimize BNNs.

To understand the existing optimization bottleneck in BNNs, we first focus on the recent popular module designs in binary architectures. In particular, researchers have proposed various performance-enhancing structures, such as double residual connection (Liu et al., 2018), as introduced in Real2BinaryNet (Martinez et al., 2020; Zhang et al., 2021), and adaptive distribution reshaping (Liu et al., 2020b). However, the shared insight behind these techniques remains unclear.

Liu et al. (Liu et al., 2021b) studied the impact of training strategies and optimizers on BNNs by visualizing the loss landscape. Inspired by this idea, we further use it as an indicator to analyze the most popular binary architectures, including BinaryNet (Courbariaux et al., 2016), BiRealNet (Liu et al., 2018), Real2BinaryNet (Martinez et al., 2020), and their full-precision counterpart ResNet-18 (He et al., 2016). To understand the relationship between binary architecture design and model optimization, We follow (Li et al., 2018) and plot the corresponding 2D (contour line view) and 3D loss landscapes for each network. The core idea behind visualization is described in the algorithm 1. Given a pre-trained BNN characterized by its filters, symbolized as $\theta$, we first project the filters in each layer into two orthogonal dimensions by combing $\theta$ with independent Gaussian noises $\sigma$ and $\eta$ as follows:

$$\mathcal{F}_n(\alpha, \beta, D) = \mathcal{L}(\theta + \alpha\sigma + \beta\eta, D), \tag{3}$$

Then the network prediction loss along with $\alpha$ and $\beta$ (ranging from -1 to 1) are recorded in Fig. 2 as a proxy observation of the high-dimensional loss landscape.

Fig.2a and Fig.2b reveal that direct binarization of the network (Courbariaux et al., 2016) creates an extremely discontinuous and rugged loss landscape surface. Such a highly rugged surface makes BNNs prone to converge into sub-optimal minima and sensitive to inappropriate optimization processes. In Fig. 2c, we observe that the double skip-connection design proposed in BiRealNet (Liu et al., 2018) alleviates the information sparsity problem to some extent and allows more continuous information flow through bypass connections. The contours are more structured and complete, although still appear rugged. Real2BinaryNet (Martinez et al., 2020) suggests reshaping the output of binary convolution in a data-driven manner using a trainable attention module, which reduces the information bottleneck. In comparison to Fig.2b and 2c, Real2BinaryNet further flattens the area around the global minimum, and the gap between each contour line in Fig.2d is more uniform, implying greater robustness to initialization and optimization.

---

**Algorithm 1** Visualization the loss landscape of BNNs

---

**Require:** A pretrained BNNs with filters $\theta$, training dataset $D$, two random sampled Gaussian direction vectors $\sigma$ and $\eta$, two scalars $\alpha$ and $\beta$.
  **for** $l$ in layers **do**:
    **for** $i$ in filters **do**:
      $\sigma_l^i \leftarrow \theta_l^i \frac{\sigma_l^i}{||\sigma_l^i||}, \quad \eta_l^i \leftarrow \theta_l^i \frac{\eta_l^i}{||\eta_l^i||},$                          ▷ filter norm Li et al. (2018)
    **end for**
  **end for**
  **for** $\alpha$ in range(-1, 1, 50) **do**:                                   ▷ direction x
    **for** $\beta$ in range(-1, 1, 50) **do**:                              ▷ direction y
      $\mathcal{F}_n(\alpha, \beta, D) = \mathcal{L}(\theta + \alpha\sigma + \beta\eta, D),$                   ▷ loss value
    **end for**
  **end for**
  Plotting the 2D nad 3D loss landscape visualization using $\mathcal{F}_n(\alpha, \beta, D)$

---

Inspired by these findings in the visualization, We have identified three core architectural design factors that influence the optimization friendliness of BNNs: (1) Activation binarization restricts the number of feature patterns and diversity per layer. (2) Proper reshaping of distribution after binary convolution is essential for effective feature modeling between adjacent 1-bit processing units. (3) Effective shortcut designs can enhance information density in forward and backward propagation. Although previous works have tried improving BNNs partially from the summarized principles above, our visualization results have demonstrated a clear optimization friendliness gap, and the need for further improvement compared to the full-precision backbone,

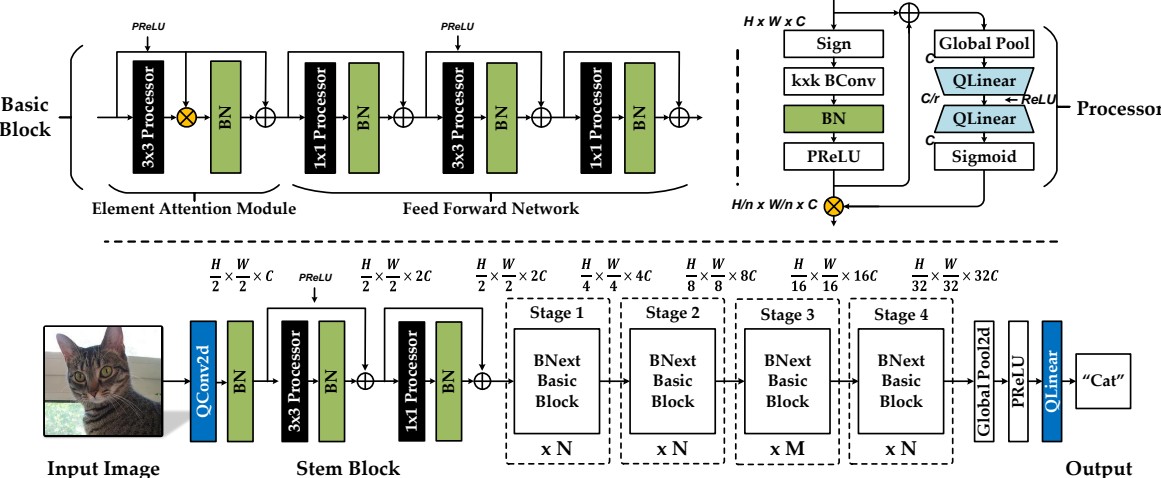

Figure 4: The architecture of BNext. "Processor" is the basic binary convolution with improved Info-RCP design, "BN" represents batch normalization. "C" indicates the base width (channel number). "N" and "M" are the number of basic blocks in each stage respectively.

which motivates us to build BNext, a strong binary baseline based on an optimization-friendly basic block design.

## 3.2 Building Optimization-Friendly Binary Neural Networks

This section introduces the core building components and design ideas of our BNext network, a binary architecture that is optimized for efficient information propagation and reduced bottlenecks.

**Binary Convolution with Information Recoupling** We propose a new binary convolution module with adaptive information reshaping and coupling to avoid bottlenecks between 1-bit processing units. Our module enhances information propagation, as shown in Figure 3c. We adjust the output distribution of a binary convolution during the forward pass using Batch Normalization and a PReLU layer, and explicitly calibrate information flow via a shift and scale design. We use a shortcut connection to merge the input and output of the main branch [BConv-BN-PReLU] by addition, and feed the output into a gap-aware Squeeze-and-Expand (SE) branch (Hu et al., 2018), which scales the main branch output. Our design, called the Information-Recoupling (Info-RCP) module, offers a more robust and flexible post-convolution reshaping mechanism than existing approaches such as XNOR-Net and Real2Binary-Net (Fig.3a and 3b). Our ablation study (Table 5) confirms the effectiveness of Info-RCP in improving network performance. Moreover, using Info-RCP results in a more uniform and smooth loss landscape (Fig. 2e), indicating that it facilitates optimization.

**Basic Block Design with ELM-Attention** We propose a novel basic block design with an enhanced bypass structure, inspired by recent attention design ideas in vision transformers and our analysis in Sec. 3.1. Our design utilizes multiple Info-RCP modules stacked together to form the fundamental building block (Processor), with each Processor followed by a BatchNorm layer. To relieve information bottlenecks in forward propagation, we enclose each basic Processor [Info-RCP-BN] using continuous residual connections. Additionally, we introduce an element-wise shortcut multiplication to dynamically calibrate the output of the first 3×3 Processor. Compared to double skip-connection design (Liu et al., 2018), this element-wise multiplication improves forward feature fusion and propagation in each block, significantly improving the loss landscape for BNNs. We demonstrate that combining both Info-RCP and ELM-Attention yields a significantly smoother loss landscape, as shown in Fig. 2g, which is for the first time comparable to the full-precision backbone. An intuitive explanation to the synergistic smoothing effect of both modules is that this two modules work as a mixed channel-wise and spatial-wise attention together in the basic block design.

Table 1: BNext architecture overview. "Q" represents the quantization bit-width of weights and activations (W/A) in Fig. 4 for the first layer, the SE branch and the output layer, respectively.

| Models | Stage Ratios (N,M) | Base Width (C) | Q (W/A) |
|---|---|---|---|
| BNext-T | 1:1:3:1 | 32 | 8/8-4/8-8/8 |
| BNext-S | 1:1:3:1 | 48 | 8/8-4/8-8/8 |
| BNext-M | 2:2:4:2 | 48 | 8/8-4/8-8/8 |
| BNext-L | 2:2:8:2 | 64 | 8/8-4/8-8/8 |

The coupled attention design largely improves the representational capacity in forward information flow and enables each basic block to recalibrate the shifted features from two orthogonal dimensions. Therefore, the basic block design implicitly rebuilds the continuous processing ability of full-precision convolutional basic block.

**BNext Family Construction** We propose the BNext family, which comprises of basic blocks stacked under different stage design strategies. To provide a fair comparison with existing works, we apply our proposed block designs to the widely-used MobileNetV1 backbone and create BNext variants by adjusting the block width and depth in each stage. Fig.4 illustrates the overall BNext family architectures, while Table 1 presents the specific configurations of the BNext networks. The width of the input layer is denoted by $C$, while $N$ and $M$ represent the stage depth ratios, as shown in Fig.4. Though the MobileNetV1 version of BNext has already shown promising improvements over previous works, We build four BNext variants with increasing capacity to assess the scalability of the BNext architecture design and previously unexplored accuracy boundary of BNNs.

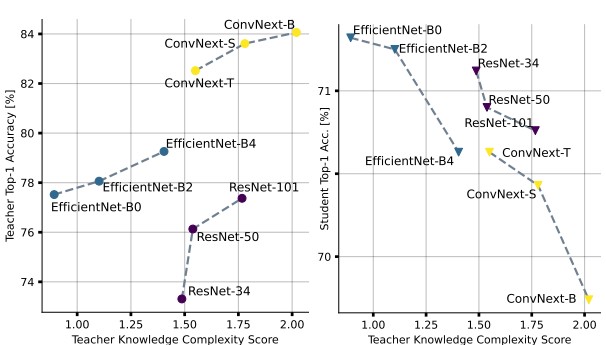

Figure 5: Relationship between binary model overfitting and the knowledge-complexity impacts under the selection of different teachers. The two y-axes indicate the teacher (left) and student (right) test Top-1 accuracy.

---

**Algorithm 2** Consecutive Knowledge Distillation

---

**Require:** A binary student $S_\theta$, a strong teacher $T_\theta^s$ and a group of assistant teachers $\{T_{\theta,1}^w, ..., T_{\theta,n}^w\}$. Training dataset $D$. Total epochs $N$. Initial assistant teacher index $m = 1$. AMP quantization $Q$, provided by PyTorch.

**for** $i$ in epochs **do**: ▷ start training

    **for** $j$, $x$ in enumerate($D$) **do**:

        1.forward $S_\theta(x), Q(T_\theta^s)(x), Q(T_{\theta,m}^w)(x)$,

        2.calculate $\mathcal{L}(x, T_\theta^s, T_\theta^w)$, Func. 5,

        3.backward and updates, Func. 1.

    **end for**

    **if** $i\%\left(\frac{N}{n}\right)=0$ **then**: ▷ switch reference teacher

        m+=1, ▷ increase knowledge effectiveness

    **end if**

**end for** ▷ end training

---

## 4 Consecutive Knowledge Distillation

Knowledge distillation (KD) is a vital optimization technique for binary neural networks (BNNs) (Martinez et al., 2020; Liu et al., 2020b; 2021b; Zhang et al., 2021). but rarely work has paid attention to the impacts and principles of teacher selections for BNNs. In this work, we propose a simple yet effective metric, *Knowledge*

*Complexity* (KC) incorporating accuracy and compactness for efficient teacher selection:

$$\mathcal{KC}(T) = \log \left( 1 + \frac{\text{Params}(T)}{\text{Accuracy}(T) + 10^{-8}} \right), \tag{4}$$

where $T$ is the selected teacher model, *Params* is the number of parameters (MB) and *Accuray* is the teacher's test top-1 accuracy. Generally speaking, KC defines the cost of teacher parameter-space complexity to the dark knowledge effectiveness.

It is commonly believed that using a stronger teacher will increase the student's accuracy upper limit. However, we empirically show in Fig. 5 that a teacher with both high accuracy and high KC can easily cause overfitting problems in binary student optimization, where the large capacity gap and prediction discrepancy between BNNs and teachers are not considered. Similar observations have been reported PokeBNN (Zhang et al., 2021), which trys using a high-precision VIT teacher instead of a ResNet-50 (Dosovitskiy et al., 2020) for binary student optimization. However, the results show that a more accurate teacher does not necessarily improve the student model's accuracy, indicating that teacher accuracy alone is insufficient for optimal performance. Instead, we suggest prioritizing teachers with lower KC for BNNs' Knowledge Distillation (KD) training.

Motivated by our observation in Fig. 5, we further develop a consecutive knowledge distillation algorithm (Alg. 2) to improve the optimization of BNNs. Specifically, we first build a self-paced knowledge ensemble to control the complexity degree of teacher knowledge $T_\theta^s$ by adding another less confident reference teacher $T_\theta^w$ into consideration. The binary student $S_\theta$ consistently learns from two teachers during the training process. The overall loss functions are mathematically formulated as follows:

$$\mathcal{L}(x) = \sigma(x) \, KL_{S_\theta, T_\theta^s}(x) + (1 - \sigma(x)) \, KL_{S_\theta, T_\theta^w}(x), \tag{5}$$

$$\sigma(x) = \frac{\exp\left(-\mathcal{L}_{CE}(S_\theta(x), T_\theta^s(x))\right)}{\sum_i \exp\left(-\mathcal{L}_{CE}(S_\theta(x), T_\theta^i(x))\right)}, \tag{6}$$

where $CE$ and $KL$ indicate the Cross-Entropy and KL-Divergence loss (Shen & Savvides, 2020), respectively. The scaling factor $\sigma(x)$ determines the relative weight of the two terms for each input example $x$. It is calculated as the softmax of the cross-entropy losses between the student model and each teacher model output distribution. Overall, the loss function aims to balance the preservation of the teacher model's capacity with the accuracy of the student model's predictions by using a reference distribution in the second term.

In order to bridge the gap in knowledge, we propose a novel knowledge-evolution strategy. During the training process, our model employs a strong teacher $T_\theta^s$, which remains consistent throughout. However, the reference teacher is selected from a pre-defined group of candidates. We introduce each reference teacher in ascending order of accuracy and knowledge complexity, allowing us to adapt our references at different stages of the training process. This approach not only lowers the optimization difficulty but also promotes better knowledge diversity as a regularization technique. Further details on the reference teacher group settings are available in the supplementary material.

## 5 Experiments

In this section, we evaluate our model and methods on the ILSVRC12 ImageNet (Deng et al., 2009) and the CIFAR.

### 5.1 Experimental Setups

We employ the AdamW optimizer (Loshchilov & Hutter, 2017) with a learning rate of $10^{-3}$ and weight decays of $10^{-3}$ and $10^{-8}$ for non-binary and binary parameters, respectively. We warm up the learning rate for 5 epochs and then reduce it using a cosine scheduler (Paszke et al., 2019). We set the backward gradient clipping range of the STE (Func.1) as [-1.5, 1.5]. We use hard binarization (Func.1) for activation

Table 2: Performance comparison with modern DNN designs and SOTA binary designs on ImageNet. "W/A" indicates the bit-width of convolutions except for the input layer. "BOPs" is the number of binary multiply-addition calculation (MAC) operations, "QOPs" is the INT-4/8 MAC operations in total, "OPs" is the total MAC, "Param" is the real parameter size, and "†" indicates results with post-training quantization on none-binary layers. In most cases, BNext performs better than models of similar (or larger) size - marked with (a) through (e).

| | Regular Designs | W/A | OPs $(10^8)$ | #Param (MB) | Top-1 (%) |
|---|---|---|---|---|---|
| (a) | LQ-Net-18 | 2/2 | - | 8.4 | 64.9 |
| | LQ-Net-18 | 4/4 | - | 16.8 | 69.3 |
| (b) | MobileNetV2 | 8/8 | 0.8 | 3.5 | 68.3 |
| (d) | ResNet-50 | 8/8 | 4.8 | 24.6 | 74.7 |
| | ResNet-18 | 32/32 | 18.0 | 44.6 | 69.6 |
| (c) | MobileNetV3 | 32/32 | 2.2 | 21.6 | 75.2 |
| | ResNet-50 | 32/32 | 38.0 | 97.5 | 76.0 |
| (e) | DeiT-S | 32/32 | 46.0 | 83.9 | 79.7 |
| (e) | RegNetY-4G | 32/32 | 40.0 | 80.1 | 80.0 |
| | Swin-T | 32/32 | 45.0 | 106.0 | 81.3 |
| | ConvNext-T | 32/32 | 45.0 | 114.3 | 82.5 |

| | Binary Designs | BOPs $(10^9)$ | QOPs $(10^6)$ | FLOPs $(10^8)$ | OPs $(10^8)$ | #Param (MB) | Top-1 (%) |
|---|---|---|---|---|---|---|---|
| | BNN | 1.70 | - | 1.20 | 1.47 | 4.2 | 42.2 |
| | XNOR-Net | 1.70 | - | 1.20 | 1.47 | 4.2 | 51.2 |
| | XNOR-Net++ | 1.70 | - | 1.20 | 1.47 | 4.2 | 57.1 |
| (a) | Bi-RealNet-18 | 1.68 | - | 1.39 | 1.65 | 4.2 | 56.4 |
| (b) | Bi-RealNet-34 | 3.53 | - | 1.39 | 1.94 | 5.1 | 62.2 |
| | Bi-RealNet-152 | 10.7 | - | 4.48 | 6.15 | - | 64.5 |
| (b) | MeliusNet-29 | 5.47 | - | 1.29 | 2.14 | 5.1 | 65.8 |
| | MeliusNet-42 | 9.69 | - | 1.74 | 3.25 | 10.1 | 69.2 |
| | MeliusNet-59 | 18.30 | - | 2.45 | 5.30 | 17.4 | 71.0 |
| (a) | Real2Binary-Net | 1.67 | - | 1.56 | 1.82 | 5.1 | 65.4 |
| (a) | ReActNet-BiR18 | 1.68 | - | 1.63 | 1.89 | 4.2 | 65.9 |
| (b) | ReActNet-A | 4.82 | - | 0.12 | 0.87 | 7.4 | 69.4 |
| (b) | ReActNet-Adam | 4.82 | - | 0.12 | 0.87 | 7.4 | 70.5 |
| (d) | PokeBNN 2.0x | 14.14 | 25.5 | - | 2.27 | 20.7 | 77.2 |
| (a) | BNext-18 † (ours) | 1.68 | 135.2 | - | 0.43 | 2.2 | **67.9** |
| (b) | BNext-T † (ours) | 4.82 | 13.4 | - | 0.77 | 5.3 | **72.0** |
| (c) | BNext-S † (ours) | 10.84 | 21.1 | - | 1.72 | 11.1 | **75.8** |
| (d) | BNext-M † (ours) | 20.09 | 24.3 | - | 3.17 | 20.4 | **77.9** |
| (e) | BNext-L † (ours) | 52.15 | 39.4 | - | 8.19 | 47.6 | **80.4** |

| Our Design w/o Post-Quant. | W/A | OPs $(10^8)$ | #Param (MB) | Top-1 (%) |
|---|---|---|---|---|
| BNext-18 (ours) | 1/1 | 1.64 | 5.4 | 68.4 |
| BNext-T (ours) | 1/1 | 0.88 | 13.3 | 72.4 |
| BNext-S (ours) | 1/1 | 1.90 | 26.7 | 76.1 |
| BNext-M (ours) | 1/1 | 3.38 | 46.5 | 78.3 |
| BNext-L (ours) | 1/1 | 8.54 | 106.1 | 80.6 |

Table 3: Design and optimization principle comparison with existing BNNs. BNext is in line with the current works but shows a higher efficiency on binary operation ratio.

| Methods | BN | 32-bit Bypass | KD | SE | BOPS/(OPS*64) |
|---|---|---|---|---|---|
| BNN Hubara et al. (2016) | ✓ | ✓ | | | 18.07% |
| BiReal-18 Liu et al. (2020a) | ✓ | ✓ | | | 17.86% |
| Real2BinaryNet Martinez et al. (2020) | ✓ | ✓ | ✓ | ✓ | 14.34% |
| ReActNet-A Liu et al. (2020b) | ✓ | ✓ | ✓ | | 86.50% |
| PokeBNN-2x Zhang et al. (2021) | ✓ | ✓ | ✓ | ✓ | 97.15% |
| BNext-18† | ✓ | ✓ | ✓ | ✓ | **61.04%** |
| BNext-T† | ✓ | ✓ | ✓ | ✓ | **97.81%** |
| BNext-L† | ✓ | ✓ | ✓ | ✓ | **99.49%** |

and progressive weight binarization as proposed in (Guo et al., 2021). For the evaluation of our technique, we use two different backbones: ResNet-18 (He et al., 2016) and ReActNet (Liu et al., 2020b). The models are trained on 8 Nvidia DGX-A100 GPUs.

**ImageNet:** We train our model with an input resolution of 224x224 and a batch size of 512 for 512 epochs. To enhance data augmentation, we use RandAugment (7, 0.5) (Cubuk et al., 2020). For the BNext optimization, we use the Diversified Consecutive KD process described in Section. 4, with ConvNext-Tiny (Liu et al., 2022) as the strong teacher and [EfficientNet-(B0-B2-B4)(Tan & Le, 2019), ConvNext-Tiny] as the candidate reference teacher groups.

**CIFAR:** We train the model with a batch size of 128 for 256 epochs, using standard data augmentations such as random crop, random horizontal flip, and normalization (Paszke et al., 2019). We optimize the model using cross-entropy loss (Paszke et al., 2019). To account for variation, we report the average of five runs.

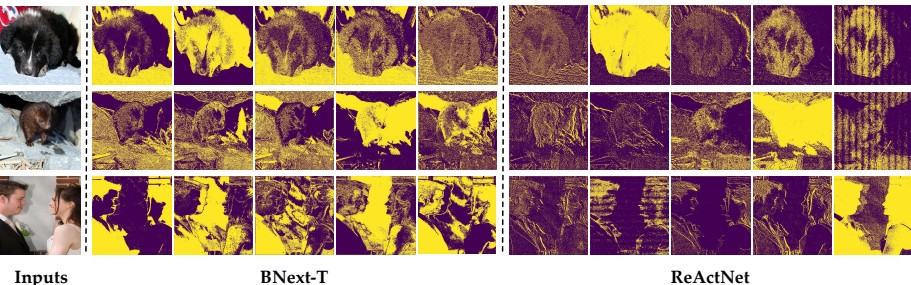

Figure 6: Binary features between BNext-T and ReActNet. BNext-T holds more diversified binary features without checkerboard artifacts compared to ReActNet, showing improved feature modeling between adjacent 1-bit processing units. Best viewed in color.

Table 4: A comparison of state-of-the-art BNNs on CIFAR10 based on the same ResNet-18 architecture as backbone.

| Method | W/A (Bitwidth) | Top-1 Acc [%] | Top-5 Acc [%] |
|---|---|---|---|
| Baseline (ResNet-18) | 32/32 | 94.8 | - |
| RAD Ding et al. (2019) | 1/1 | 90.5 | - |
| IR-Net Qin et al. (2020) | 1/1 | 91.5 | - |
| RBNN Lin et al. (2020) | 1/1 | 92.2 | - |
| ReCU Xu et al. (2021) | 1/1 | 92.8 | - |
| AdaBNN Tu et al. (2022) | 1/1 | 93.1 | - |
| BNext (ResNet-18) (ours) | 1/1 | **93.6** ($\pm$0.12) | 98.8 ($\pm$0.03) |

## 5.2 Performance Evaluation

**ImageNet** We evaluate the performance of the proposed BNext family by comparing it to existing designs under different capacity levels, demonstrating the superiority of our methods. To illustrate, we compare BNext-Tiny† to the popular ReActNet (Liu et al., 2020b) design, where BNext-Tiny† achieves 2.6% higher accuracy with 10M fewer operations. Moreover, our BNext-M† outperforms the previous state-of-the-art design, PokeBNN-2.0x (ResNet-50 backbone), by 0.7% in accuracy, with a similar model size. Notably, BNext requires lower optimization requirements than the PokeBNN family. For instance, PokeBNN is trained with an expensive batch size of 8192 for 720 epochs (Zhang et al., 2021) on 64 TPU-v3 chips, while BNext only uses a batch size of 512 for 512 epochs. To further evaluate the effectiveness of the BNext design, we combine our proposed techniques with a ResNet-18 backbone. We apply standard data augmentation and use only ResNet-34 as a teacher for a fair comparison. BNext-18 achieves a top-1 accuracy of 68.4%, which is only 1.2% lower than the original ResNet-18 (He et al., 2016). This performance surpasses other designs, such as BNN (Courbariaux et al., 2016), XNOR-Net (Rastegari et al., 2016), BiReal-Net (Liu et al., 2018), Real2BinaryNet (Martinez et al., 2020), and ReActNet-BIR18 (Liu et al., 2020b). Finally, we try to contribute a previously unexplored strong baseline BNext-L on ImageNet. For BNext-L, we do not intend to design an optimal binary architecture with the best accuracy and efficiency trade-off, but to contribute a practical baseline for future work of high-accuracy BNNs research. With the introduction of BNext-L, we have been able to achieve an impressive upper boundary for BNN performance on ImageNet, reaching a crucial accuracy level of 80.57%. With this result BNNs show a closer upper boundary to recent SOTA 32-bit designs, such as ConvNext (Liu et al., 2022), Swin Transformer (Liu et al., 2021a) and RegNetY-4G (Radosavovic et al., 2020). Meanwhile, we have given a more detailed comparison to existing BNNs in Table. 3, where BNext family shows a higher binary ratio from the aspects of binary operation ratio. For the sake of fair comparison, our calculation of OPs excludes negligible floating-point counts as (Liu et al., 2018; Bethge et al., 2020; Liu et al., 2020b; 2021b; Zhang et al., 2021). The additional training information and corresponding optimization curves of each model are available in Appendix, restricted by page limitation.

**CIFAR:** We further explore the generalization ability of BNext (with a ResNet-18 backbone) on the smaller dataset CIFAR10 (see Table. 4). Compared to the latest designs such as AdaBNN (Tu et al., 2022), ReCU

Table 5: The ablation study of Info-RCP and ELM-Attention on ImageNet, showing the Top-1/Top-5 accuracy, model size ("Param"), and total operations ("OPs").

| Methods | Top-1 (%) | Top-5 (%) | Param (MB) | OPs ($10^8$) |
|---|---|---|---|---|
| | w/o ELM-Attention | | | |
| w/o Info-Recoupling | 62.43 | 83.75 | 4.52 | 0.76 |
| w/ Info-Recoupling | 64.91 | 85.52 | 5.58 | 0.77 |
| | w/ ELM-Attention | | | |
| w/o Info-Recoupling | 64.61 | 85.49 | 4.52 | 0.76 |
| w/ Info-Recoupling | 65.02 | 85.73 | 5.58 | 0.77 |

Table 6: The ablation study of ELM-Attention position in each basic block on CIFAR100 dataset, showing the averaged Top-1 accuracy after 5 runs.

| Backbone | BNext-Tiny | | | |
|---|---|---|---|---|
| ELM-Attention Position | 1th BConv 3x3 | 1th BConv 1x1 | 2th BConv 3x3 | 2th BConv 1x1 |
| Top-1 Acc (%) | 66.88 | 66.73 | 66.34 | 66.76 |
| Backbone | BNext-18 | | | |
| ELM-Attention Position | 1th BConv 3x3 | 2th BConv 3x3 | 1/2th BConv 3x3 | w/o |
| Top-1 Acc (%) | 72.22 | 71.72 | 70.00 | 66.95 |

(Xu et al., 2021) and RBNN (Lin et al., 2020), BNext achieves a better performance. It surpasses one of the most recent works AdaBNN by 0.5% and closes the gap to 32-bit ResNet to 0.8%. The complete evaluation of CIFAR100 is provided in the supplementary material.

**On-Hardware Evaluation**: Our study evaluates inference efficiency of binary neural networks (BNNs) on a Banana Pi M5, using the Larq Library. Despite Larq's limitations—supporting only TensorFlow, we integrated it with PyTorch via the C++ extension API. In our tests, a 1-bit convolution layer with an NCHW configuration of (16, 64, 128, 128) on the Banana Pi M5 showcased a 20.1x speedup compared to native PyTorch CPU implementation. We incorporated this setup with the BNext-18 model, maintaining channel constraints from Larq, and set input dimensions to [1, 3, 224, 224]. Inference times for pure PyTorch, C++, and ARM instruction sets were 1.27s, 0.8948s, and 0.5759s respectively. The Larq integration yielded a 2.2x overall acceleration. However, as highlighted in our Limitations section, the absence of efficient BNN acceleration methods and the reliance on PyTorch's C++ API, which doesn't mitigate additional floating-point operations, may not fully showcase BNNs' acceleration potential.

### 5.3 Ablation Study

We present a comprehensive evaluation of the proposed techniques in this paper through a detailed ablation study on module designs, optimization schemes, data augmentation strategies, and post-quantization impacts. All experiments are conducted using BNext-Tiny as the base model with standard data augmentation and cross-entropy loss unless otherwise stated.

**Module Designs.** We evaluate the proposed Info-RCP and ELM-Attention modules through an ablation study, as shown in Table 5. The baseline model without these structures achieves only 62.43% accuracy on ImageNet. Adding ELM-Attention or Info-RCP to BNext-Tiny results in accuracy gains of 2.48% and 2.18%, respectively. Combining both designs achieves the highest accuracy of 65.02%, confirming that both methods enhance the model's representative capacity. Meanwhile, we futher report the corresponding overhead after using each sub-design. Both are highly lightweight and only contribute a little extra theoretical computation cost. We are also interested in the impacts of making a difference where the ELM-Attention module is used. To make it clear, we conducted an external evaluation using the BNext-18 and BNext-Tiny backbone on CIFAR100 datasets, Table 6. Applying the ELM-Attention design to the first convolution module shows better modeling ability on both backbones. This shapes the architecture similar to the popular transformer block. Either moving the ELM-Attention module to another position or removing this design

Table 7: The ablation study on the effects of adding different training strategies to BNext-T on ImageNet dataset.

| Method | Epochs | Top-1 [%] | Top-5 [%] |
|---|---|---|---|
| + KD | 128 | 70.76 | 89.64 |
| + KC-Based $T(x)$ Selection | 128 | 71.33 (+0.57) | 90.02 (+0.38) |
| + Consecutive KD | 128 | 71.38 (+0.05) | 90.04 (+0.02) |
| + Rand Augmentation Cubuk et al. (2020) | 128 | 71.46 (+0.08) | 90.06 (+0.02) |
| + Long Training | 512 | 72.40 (+0.94) | 90.64 (+0.58) |

Table 8: The ablation study on popular data augmentation strategies on ImageNet. BNext-Tiny is selected as the baseline.

| Data Augmentation | Top-1 (%) | Top-5 (%) |
|---|---|---|
| Baseline (BNext-Tiny) | 65.02 | 85.73 |
| Mixup Zhang et al. (2017) | 65.83 (+0.81) | 86.16 (+0.43) |
| Cutmix Yun et al. (2019) | 64.51 (-0.51) | 85.32 (-0.41) |
| Repeat Augment Hoffer et al. (2020) | 53.74 (-11.28) | 75.65 (-10.08) |
| Rand Augment Cubuk et al. (2020) | 67.51 (+2.49) | 87.28 (+1.55) |
| Rand Augment Cubuk et al. (2020), Mixup Zhang et al. (2017) | 66.51 (+1.49) | 86.77 (+1.04) |

gives a suboptimal performance. A deeper exploration of the attention mechanism could further improve BNN design, which is left for future work.

**Optimization Schemes.** We evaluate the effectiveness of all optimization schemes used in our experiments in Table 7. We replace the ResNet-101 teacher with a teacher selected using knowledge complexity (Efficient-B0), which results in a 0.57% accuracy gain. We further use a gap-sensitive ensemble of Efficient-B0 and Efficient-B2 and add Rand Augmentation, resulting in a total accuracy gain of 0.13%. Though combining consecutive KD with BNext-T only contributes 0.05% improvements, this can be explained by the relatively diminutive knowledge complexity gap (KC: 0.893 vs 1.103) between the two teachers. However, it's imperative to spotlight that the Consecutive KD, despite its marginal numerical impact in Table 6, has underscored a pivotal revelation on mitigating the counter-intuitive overfitting issue as portrayed in Table 9 and Figure 5. Extending the training epochs to 512 further improves the accuracy to 72.40%.

**Data Augmentations.**

Modern high-performance vision models have been benefiting from well-designed data augmentation strategies. However, the adaptation relationship to BNN has still not been effectively explored in the current research community. We evaluate the effectiveness of popular data augmentations such as Mixup, Cutmix, Augment-Repeat, and Rand Augmentation, as shown in Table 8. Most of these choices are actually harmful to BNext model optimization. For example, using the popular Augment-Repeat can decrease the Top-1 accuracy by 10.08%. Applying only Rand Augmentation achieves the best improvement of 2.49%. The adaption differences here showcase an interesting optimization gap between BNN and their 32-bit counterparts.

## 6 Conclusion

In this paper, we propose a novel optimization pipeline to enhance the optimization-friendliness of Binary Neural Networks. This is achieved by first building a strong binary baseline, BNext from the aspects of smoother loss landscape and improved optimization friendliness. Then we reconsider the vital knowledge distillation optimization and the impacts of teacher selection in binary network optimization. A simple yet effective teacher-selection metric called knowledge complexity, and a novel multi-round optimization technique, Consecutive Knowledge Distillation (CKD), is proposed to alleviate the high-confidence knowledge distillation bottleneck for BNNs. The highly accurate results on ImageNet demonstrate that improved optimization friendliness plays an important role in building high-accuracy binary neural networks.

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

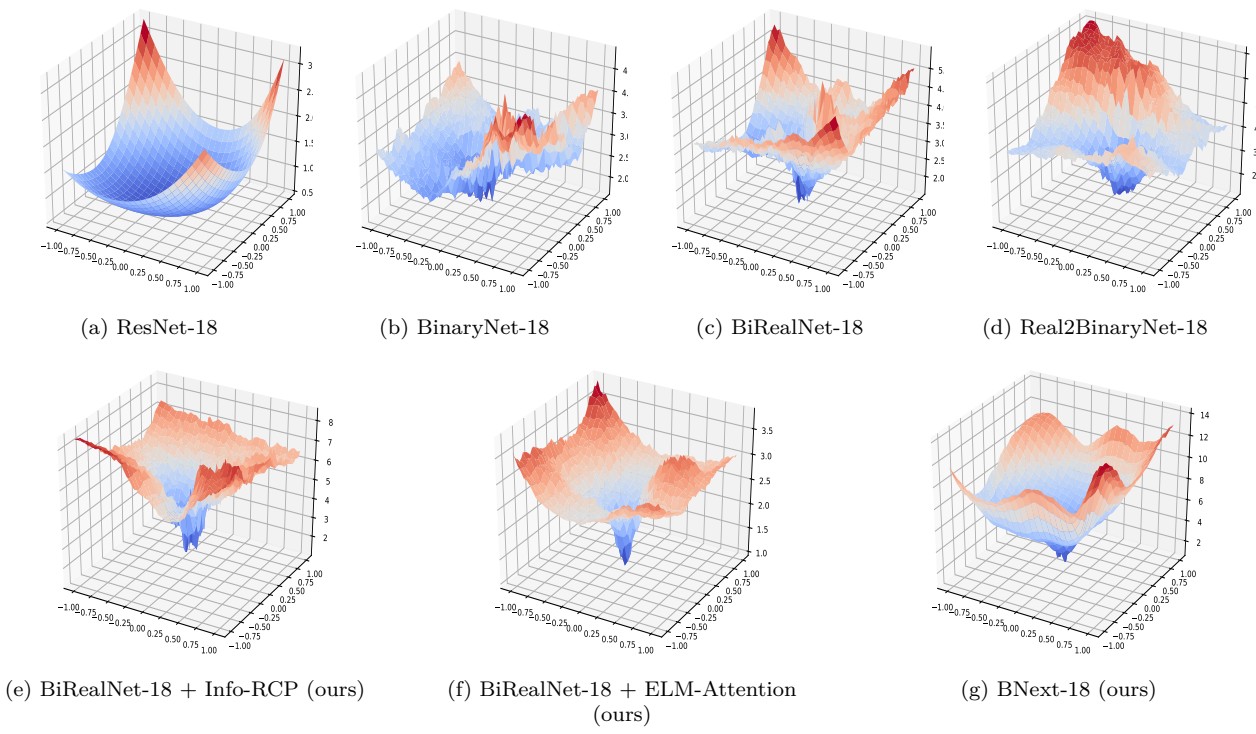

(a) ResNet-18   (b) BinaryNet-18   (c) BiRealNet-18   (d) Real2BinaryNet-18

(e) BiRealNet-18 + Info-RCP (ours)   (f) BiRealNet-18 + ELM-Attention (ours)   (g) BNext-18 (ours)

Figure 7: The loss landscape of popular BNNs (3D view) on the CIFAR10 dataset. We follow the setting of (Li et al., 2018). "Info-RCP" and "ELM-Attention" are the module designs in this paper. Compared with previous design, the BNext-18 shows a smoother loss landscape. (The corresponding contour lines are shown in Figure 2 in the main work.)

## 7    Appendix

In this section, we present more detailed visualization results and an ablation study, which are not listed in the main paper due to limited space.

### 7.1    3D Loss Landscape Visualization

We use the loss landscape visualization technique on existing binary architecture designs such as BinaryNet (Courbariaux et al., 2016), BiRealNet (Liu et al., 2018), Real2BinaryNet (Martinez et al., 2020), and their full-precision counterpart ResNet-18 (He et al., 2016). We plotted the corresponding 3D loss landscapes for each network, as shown in Fig. 7. It also shows the 3D view of the designed modules (Info-RCP and ELM-Attention) individually and our full BNext design (using both modules). It is easy to see that binarization makes the loss landscape surface really rugged but our design can alleviate this problem of binarization. As a result, the loss landscape surface of BNext is already close to the previous full precision design. This explains why BNext can reach a higher accuracy boundary.

### 7.2    Counter-Intuitive Overfitting

As explained in the main work, knowledge distillation has long been an essential choice for optimizing binary neural networks. When we move our optimization target from 70%-level to 80%-level binary neural network, it is inevitable to seek help from higher accuracy pretrained deep neural networks. More specifically, we need to decide which model is most suitable as the teacher for the optimization of highly-accuracy BNNs, such as, BNext. We choose the teacher empirically from a few popular deep neural network families, such as, ResNet, EffcientNet, and ConvNext. During this process, we observe that the BNext design can easily overfit

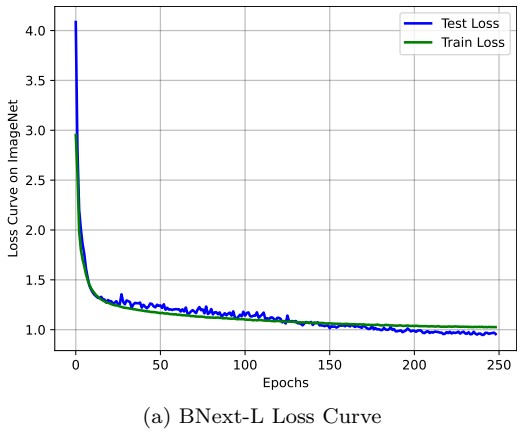

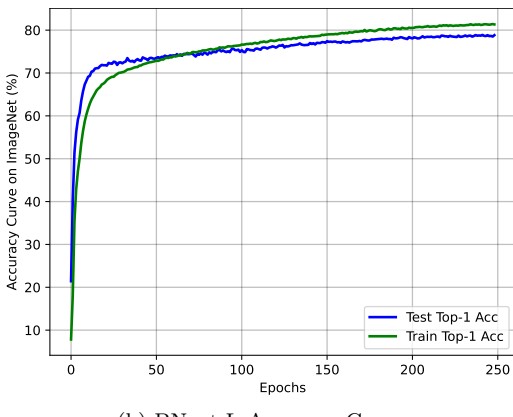

(a) BNext-L Loss Curve                    (b) BNext-L Accuracy Curve

Figure 8: The optimization procedure of BNext-L model on ImageNet dataset with standard knowledge distillation technique and ConvNext-T as teacher. During the early stage of the optimization, the student always has a lower training loss than testing loss. This indicates a better fitting to training set than testing set.

to strong and highly accurate teachers. In Table. 9, we can see that when a highly accurate full precision model as the teacher leads to a high training accuracy of the student model, but does not generalize well and even performs worse on validation data. For example, when ConvNext-Base is used as the teacher for BNext-Tiny, the student training accuracy is 3% higher than using EfficientNet-B0 as the teacher, but the testing accuracy is 1.58% lower.

For larger students such as BNext-M/L, this overfitting phenomenon still exists even with increased model capacity. Take our largest model BNext-L as an example: simply combing standard knowledge distillation with a strong teacher ConvNext-T results in a student with sub-optimal generalization. As we can see from the optimization procedure in Fig. 8, the testing loss is always higher than the training loss during the early stage of training procedure. Meanwhile, the performance on the test set is less stable compared to the training curve. When reaching the same testing accuracy of 78%, the training accuracy under standard KD is already 80.23%, which is almost 4% higher than the final version with the proposed diversified consecutive KD. After a 256 epochs training, the evaluation result on the test set is 1.67% lower than the diversified consecutive KD results in Table. 2. The overfitting shown in Fig. 8 verifies the need of redesigning the optimization pipeline for BNext.

### 7.3 Detailed Teacher Selection for BNext Family Optimization

To solve the counter intuitive overfitting problems for BNext family optimization, we designed the diversified consecutive KD in Sec. 4. For each BNext design, we build the corresponding knowledge matrix based on a comprehensive consideration of student capacity, teacher knowledge complexity and teacher evaluation performance. The detailed teacher selections are shown in Table. 10. For BNext-18, we only use the standard KD setting for a fair comparison, which means that only ResNet-34 is utilized during the training procedure. For BNext-T/S, we apply a gap-aware ensemble strategy for knowledge distillation (Sec. 4) to increase the diversity of the teacher knowledge. Since the capacity of BNext-T/S performance is relatively limited, the EfficientNet-B0/B2 models are already sufficiently strong as teachers. Consequently, no knowledge boosting strategy is utilized for them. For BNext-M/L, we rely on higher accuracy teachers like ConvNext-T to explore the accuracy boundary of our BNext design. As we have observed, simply learning from this kind of high accuracy teacher from scratch under standard KD can suffer from overfitting problem. Consequently, we use the knowledge-boosting strategy (proposed in Sec. 4) to evolve the supervision information during the training. In this way, the knowledge discrepancy between student and teacher can be decreased by slowly increasing the knowledge confidence during the training.

Table 9: The counter-intuitive overfitting problem observed in the BNext model optimization under different teacher selections. The BNext-Tiny is used for evaluation. Each model is optimized for 128 epochs using the standard data augmentation and standard knowledge distillation technique. "KC" indicates the knowledge-complexity score, Func. 4

| Teachers | KC | Teacher Test Top-1 (%) | Student Train Top-1 (%) | Student Test Top-1 (%) |
|---|---|---|---|---|
| None | 0 | 0 | 76.61 | 65.02 |
| ResNet-34 | 1.487 | 73.31 | 72.82 | 71.12 |
| ResNet-50 | 1.539 | 76.13 | 71.59 | 70.90 |
| ResNet-101 | 1.768 | 77.37 | 72.38 | 70.76 |
| EfficientNet-B0 | 0.893 | 77.53 | 72.16 | 71.32 |
| EfficientNet-B2 | 1.103 | 78.06 | 71.91 | 71.25 |
| EfficientNet-B4 | 1.405 | 79.26 | 73.88 | 70.63 |
| ConvNext-T | 1.552 | 82.52 | 73.63 | 70.63 |
| ConvNext-S | 1.786 | 83.66 | 74.29 | 70.43 |
| ConvNext-B | 2.027 | 84.06 | 75.22 | 69.74 |
| ResNext101-32x8D | 2.053 | 79.32 | 75.42 | 69.67 |

Table 10: The detailed teacher settings for our BNext model optimization.

| Students | Strong Teacher | Assistant Teacher Groups |
|---|---|---|
| BNext-18 | ResNet-34 | None |
| BNext-T | Efficient-B2 | Efficient-B0 |
| BNext-S | Efficient-B2 | Efficient-B0 |
| BNext-M | Efficient-B4 | Efficient-B0, Efficient-B2 |
| BNext-L | ConvNext-T | Efficient-B0, Efficient-B4, ConvNext-T |

## 7.4 BNext Model Training Procedure

The detailed training procedure of the BNext family (T/S/M/L) are shown in Fig. 10. Since each BNext design holds different teacher matrix and variant teacher scheduler, we can observe different patterns in the training loss curve and training accuracy curve of each model. All in all, we can see that each BNext model always has a lower testing loss than training loss during the optimization. If we further compare the testing curves between Fig. 10j and Fig. 8a, we can conclude that the proposed diversified consecutive KD helps BNext optimization to generalize better on testing set.

## 7.5 BNext Post-Quantization Details

We try to maximize the efficiency of BNext family by further quantizing the input layer, the last layer and the Squeeze-and-Expand branch in each Info-RCP module. Specifically, we utilize post-training quantization for each BNext model. For the optimization settings, we use the AdamW optimizer ($\beta_1$=0.99, $\beta_2$=0.999) to fine-tune the corresponding layers but keep the pretrained binary layers fixed. The initial learning rate is 1e-7 and each model is fine-tuned for 5 epochs on the ImageNet dataset. The data augmentation is kept in line with the pre-training phase. No weight decay is used during this period. We use asymmetric quantization,

Table 11: 5 runs evaluation of BNext-18 on the CIFAR10 and CIFAR100 datasets.

| Datasets | Train Top-1 (%) | Train Top-5 (%) | Test Top-1 (%) | Test Top-5 (%) |
|----------|-----------------|-----------------|----------------|----------------|
| CIFAR10-0 | 99.99 | 100 | 93.67 | 99.86 |
| CIFAR10-1 | 99.99 | 100 | 93.39 | 99.79 |
| CIFAR10-2 | 99.99 | 100 | 93.59 | 99.83 |
| CIFAR10-3 | 99.99 | 100 | 93.69 | 99.84 |
| CIFAR10-4 | 99.99 | 100 | 93.66 | 99.81 |
| Avg | 99.99 | 100 | 93.60 | 98.83 |
| Var | 0.00 | 0.00 | 0.0152 | 0.0007 |
| Std | 0.00 | 0.00 | 0.1233 | 0.027 |
| CIFAR100-0 | 99.97 | 100 | 72.22 | 91.23 |
| CIFAR100-1 | 99.98 | 100 | 72.02 | 91.75 |
| CIFAR100-2 | 99.97 | 100 | 72.35 | 91.61 |
| CIFAR100-3 | 99.97 | 100 | 71.97 | 91.51 |
| CIFAR100-4 | 99.97 | 100 | 72.37 | 91.64 |
| Avg | 99.97 | 100 | 72.18 | 91.55 |
| Var | 2e-5 | 0.00 | 0.032 | 0.0379 |
| Std | 0.0045 | 0.00 | 0.1845 | 0.1945 |

which can be mathematically formulated as follows:

$$X_q^i = \frac{\text{round}((X_r^i - \beta) \cdot \alpha)}{\alpha} + \beta, \tag{7}$$

$$\alpha = \frac{2^n - 1}{\max_{X_r^i} - \min_{X_r^i}}, \quad \beta = \min_{X_r^i}, \tag{8}$$

$$W_q^i = \left( \frac{\text{round}((W_r^{i,\text{Scaled}}) - \beta) \cdot \alpha}{\alpha} + \beta \right) \cdot \text{Norm}(W_r^i), \tag{9}$$

$$\alpha = \frac{2^n - 1}{\max_{W_r^{i,\text{Scaled}}} - \min_{W_r^{i,\text{Scaled}}}}, \quad \beta = \min_{W_r^{i,\text{Scaled}}}. \tag{10}$$

where $X^i$ represents the input features and $W^i$ represents weights respectively. The $r$ and $q$ means full precision and quantization representation. The $Norm$ indicates the absolute-mean for each output channel. The $Scaled$ means that the variable is scaled down by its channel-wise absolute-mean.

### 7.6 Detailed CIFAR Evaluation Results

As we have mentioned in the main pages, we train a BNext-18 model on the CIFAR dataset (including CIFAR10 and CIFAR100) to evaluate the generalization of BNext design. Due to limited space, we only show the averaged results in the main page. Here, we present the detailed results of 5 runs on CIFAR, as shown in Table. 11.

### 7.7 Quantization Impacts.

We evaluate the impact of post-training quantization on the input convolution layer, output fully-connected layer, and layers in each SE branch, as shown in Table 12. Taking BNext-L as an example, We find that applying 8-bit quantization to weights and activations of all layers only marginally reduces accuracy to 80.47%. Further, decreasing the bit-width of weights in the SE branch to 4 bits (with 8-bit activations) results in a final BNext-L model with 80.37% Top-1 accuracy.

Table 12: Impact of post-training quantization on the first input layer, the last layer, and the SE branch in BNext Design. The best and second-best quantization results are highlighted.

| I-S-E-O (W/A) | BNext-18 | BNext-T | BNext-S | BNext-M | BNext-L |
|---|---|---|---|---|---|
| 32/32 | 68.37 | 72.36 | 76.06 | 78.27 | 80.57 |
| 8/8-8/8-8/8-8/8 | **68.38** | **72.36** | **76.05** | **78.11** | **80.47** |
| 8/8-4/8-4/8-8/8 | **67.94** | **72.04** | **75.75** | **77.97** | **80.37** |
| 8/8-4/8-4/4-8/8 | 67.91 | 71.95 | 75.61 | 77.93 | 80.26 |
| 8/8-4/4-4/4-8/8 | 67.29 | 71.72 | 75.50 | 77.79 | 79.52 |

## 7.8 Code Submission

Within this ZIP file we provide you with our training code, so you are able to reproduce our results if desired. We are planning to release the code and pre-trained models via Github together with the final version of this paper. (Licensing is planned with the "Apache License 2.0".) Until then, please treat this code confidentially.

We added all the details needed to reproduce each of our BNext models depicted in Section 3 of our paper in the respective folders:

- **BNext-T:** `src/script/BNext-Tiny`

- **BNext-S:** `src/script/BNext-Small`

- **BNext-M:** `src/script/BNext-Middle`

- **BNext-L:** `src/script/BNext-Large`

The complete code used for each run can be found in the subfolder `src` and the exact running command is saved in `src/script`. We also added the output logs (`logs/training.log`).

We use a virtualized environment for PyTorch (Paszke et al., 2019) based on Ananconda for our code setup. The hosts system thus needs support for Python 3.9.13 and a recent NVIDIA CUDA driver (we tested driver version 470.82.01 with CUDA 11.4 before this submission) for training with GPU.

Note that the ImageNet dataset also needs to be downloaded and prepared manually in the usual manner (using a *train* and *val* folder for the respective split). The validation images need to be moved into labeled subfolders.

## 7.9 Limitations

While the BNext design demonstrates a superior binary computation ratio compared to most preceding Binary Neural Network (BNN) architectures (Table. 3), it hosts several 4-bit linear layers within each Infor-RCP module, thereby introducing additional complexity. A promising avenue for addressing this issue is refining the design of the Infor-RCP module and bolstering the optimization of the binary components, which will be explored as part of our future work. Although the employed consecutive knowledge distillation technique has provided valuable guidance for binary model optimization, the substantial GPU memory demands necessitated by maintaining multiple teacher models hampers its user-friendliness. An effective remedy could encompass evolving the existing mix-precision teacher models towards lower bit representations such as INT8 or even 4-bit configurations. This transition is expected to markedly diminish the computational load during the training phase and subsequently alleviate the knowledge complexity for student models. The anticipated speed-up, as posited theoretically, awaits validation on actual hardware. Given the absence of a proficient GPU implementation for any BNN at present, efforts have been initiated to develop a GPU-accelerated version of BNext. However, as of this paper's submission, the outcomes of these endeavors remain forthcoming.

The presented work, while substantial in its contributions towards accuracy enhancement and theoretical acceleration aligned with preceding efforts, holds potential for further advancements. We are optimistic

about furnishing an open-source GPU implementation alongside inference speed verification results in the subsequent phases of this research, thereby propelling the practical applicability and performance verification of the BNext design.

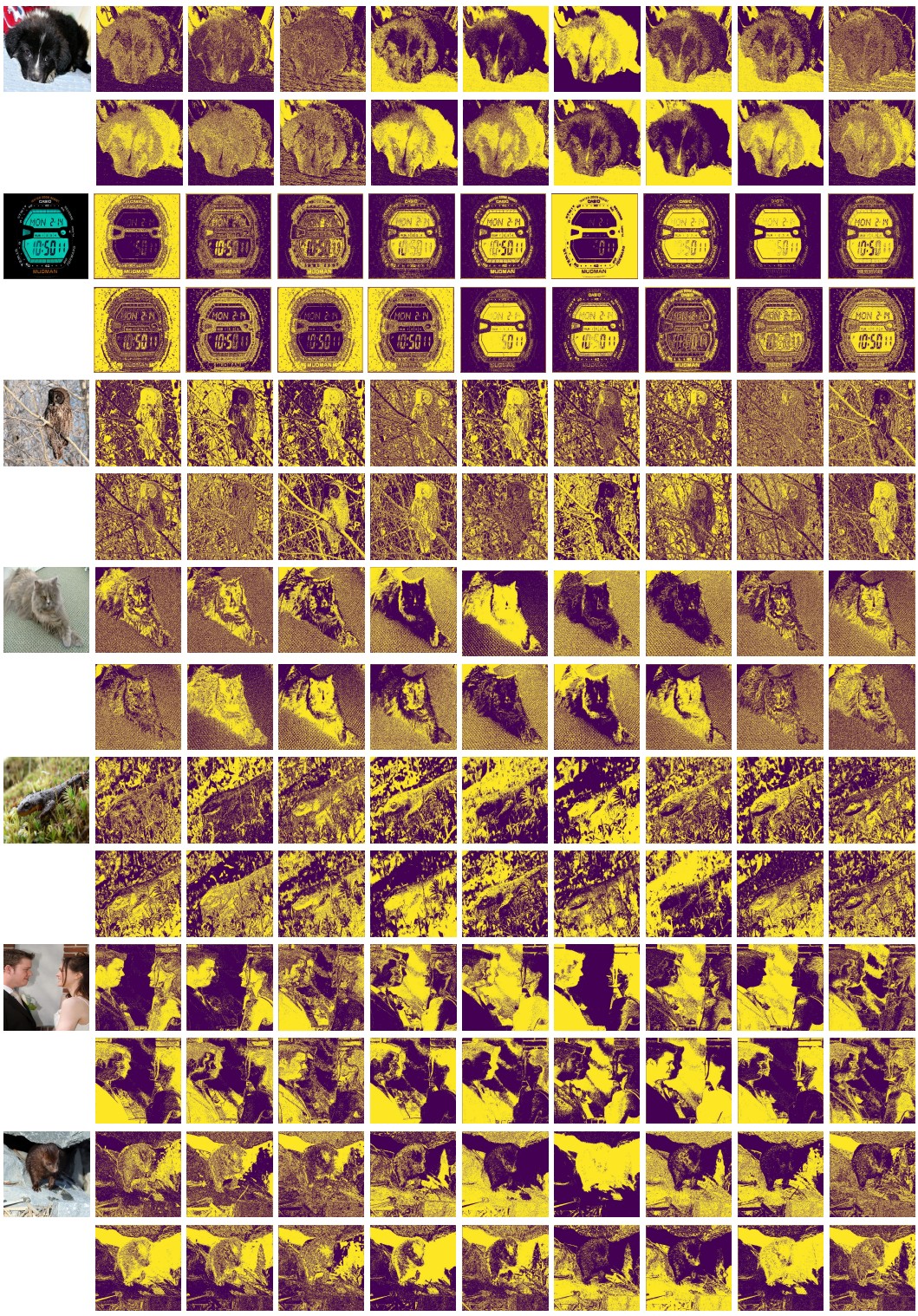

Figure 9: The binary feature map diversity visualization for BNext-T. We select the binary inputs of the first binary convolution in stage 1 for visualization. The input images are resized to 1024x1024 for better comparison.

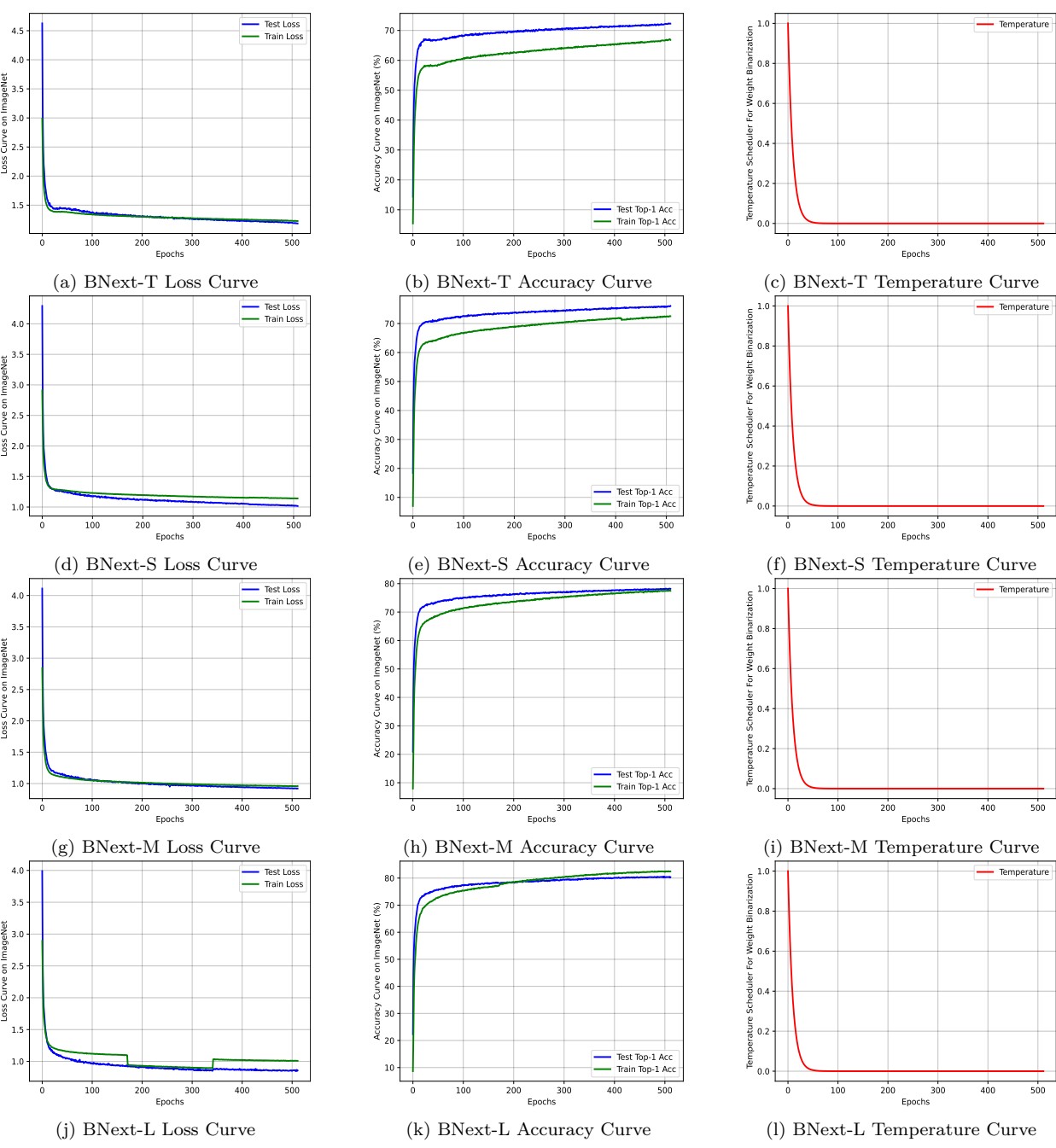

(a) BNext-T Loss Curve     (b) BNext-T Accuracy Curve     (c) BNext-T Temperature Curve

(d) BNext-S Loss Curve     (e) BNext-S Accuracy Curve     (f) BNext-S Temperature Curve

(g) BNext-M Loss Curve     (h) BNext-M Accuracy Curve     (i) BNext-M Temperature Curve

(j) BNext-L Loss Curve     (k) BNext-L Accuracy Curve     (l) BNext-L Temperature Curve

Figure 10: The training procedure of BNext family on ImageNet dataset. The temperature curve indicates the temperature scheduler for the progressive weights binarization.

