# OpenReview forum: "Towards Optimization-Friendly Binary Neural Network"
_TMLR — Accepted by TMLR_

### Review · Reviewer_FZQj · 2023-10-04

**Summary Of Contributions:**

The author proposed a novel optimization-friendly architecture based on their understanding of the loss landscape of models. They also proposed a metric for picking the proper teacher model for training student binary models. A SOTA model accuracy is achieved with their approach.

**Audience:**

Yes

**Broader Impact Concerns:**

N.A.

**Claims And Evidence:**

No

**Requested Changes:**

1. Suggest to include a short summary of how the loss landscape is visualized so readers can avoid looking it up as it is a core part of the study .
2. P6. Please fix (Fig.?? and ??).
3. P6. Please fix  designLiu
4. The architecture of BNext is not clearly illustrated in Figure 4. Not sure about the Basic Block part of Figure 4, if the multiplication operation of the first segment is performed twice.
5. Please clearly define OPs, BOPs, QOPs, and the meaning of #Param (MB)
6. Table 4. without the context of network architectures of the models in comparison, it is difficult to tell if the comparison is fair.
7. Table 7: it is not clear what the baseline is.
8. Model distillation: it is difficult to tell if the result is due to a specific choice of the distillation hyper-parameters.
9. It is difficult to tell if the proposed improvement is due to model architecture, augmentation strategy, distillation, or more extended training.

**Strengths And Weaknesses:**

Strength:
Achieved improved model performance. Analysis with loss landscape provide a window into understanding BNN optimization.
Weakness:
Presentation should be improve, please see below.

---

> ### Author Response · Authors · 2023-10-31
> **Response to Reviewer FZQj`s Comments on Paper1603**
>
> First and foremost, we sincerely appreciate the reviewer's detailed analysis of our paper, especially in terms of the performance improvements and the deep insights our analysis provides into BNN optimization. We also recognize the presentation weaknesses and are committed to addressing them.
>
> ### 1. Loss Landscape Visualization Summary:
> **Response**:We have incorporated a brief summary and algorithm pipeline elucidating the methodology employed for visualizing the loss landscape in methodology parts. This addition aims to offer readers a direct insight into the core aspect of our study without necessitating external lookup.
>
> ### 2.Figure and Text Corrections (P6).
> **Response**:We have rectified the placeholder references in the text (Fig.?? and ??) and the term "designLiu" to ensure accurate referencing and coherence.
>
> ### 3. Architecture Illustration (Figure 4).
> **Response**:We have refined Figure 4 to depict the architecture of BNext more clearly. Additionally, there is no recurrent design in the Basic Block part. Therefore, the multiplication operation of the first segment will not be performed twice.
>
> ### 4. Definition of OPs, BOPs, QOPs, and #Param (MB).
> **Response**: We have provided clear definitions and explanations in the revised version, for OPs (Total Multiply and Addition Operations), BOPs (Binary Multiply and Addition Operations), QOPs (Quantized Multiply and Addition Operations), and #Param (MB) (Number of Parameters in Megabytes) within the text to elucidate the terminology used.
>
> ### 5. Contextual Information for Table 4.
> **Response**: To enable a fair comparison, we have provided more contextual information regarding the network architectures of the models being compared in Table 4. To be in line with previous work and for a fair comparison, all the comparison on CIFAR is based on the ResNet18 backbone.
>
> ### 6. Baseline Clarification (Table 7).
> **Response**: We are sorry for the unclear description. As claimed in the table caption, the BNext-Tiny is selected as the baseline. We have reorganized Table 7 to aid readers in comprehending the comparative analysis presented.
>
> ### 7. Influence of specific choice of the distillation hyper-parameters.
> **Response**: We are sorry for the confusion of this part. We understand the concern that a specific choice of the distillation hyper-parameters could influence the result. However, we wanna mention that this work is the first to explore the knowledge distillation influence for high-accuracy (80% +) BNN optimization. For the observation of the counter-intuitive overfitting problem in BNN knowledge distillation, we follow the existing setting in BNN knowledge distillation work, and all teachers are evaluated under the same and fair hyper-parameters setting (batch size, learning rate, epochs, optimizer, learning rate scheduler). Therefore, the observation could show the characteristics of teacher influences in BNN distillation, which is also one of the main contributions of this work. The consecutive distillation among different BNext families also follows an inline hyper-parameter setting and more details can be found in the provided code in supplementary materials.
>
> ### 8. Source of Proposed Improvement.
> **Response**: We are sorry for the unclear about the source of the proposed improvement. As we have analyzed in the detailed ablation study in Table 5, Table 6, and Table 7. The architecture design and proposed distillation method work together to shape the smooth optimization (which is the main motivation in this paper) of BNexts. Extra strategies such as data augmentation and long training are already common choices in modern high-performance vision models (VIT [1], ConvNext [2]) optimization and the existing SOTA works like PokeBNN [3] and ReActNet [4]. In Table 5 and Table 6, we have also given a clear ablation study on the improvement of each strategy, taking BNext-T as the backbone.
>
> **In conclusion**, we sincerely appreciate Reviewer FZQj for the constructive feedback and astute observations provided on Paper1603. By addressing each of the points you've highlighted, we believe we have enriched the content, ensuring readers gain a more comprehensive understanding of our work.
>
> **References:**
> [1] Dosovitskiy, Alexey, et al. "An image is worth 16x16 words: Transformers for image recognition at scale." arXiv preprint arXiv:2010.11929 (2020).
> [2] Liu, Zhuang, et al. "A convnet for the 2020s." Proceedings of the IEEE/CVF conference on computer vision and pattern recognition. 2022.
> [3] Zhang, Yichi, Zhiru Zhang, and Lukasz Lew. "Pokebnn: A binary pursuit of lightweight accuracy." Proceedings of the IEEE/CVF Conference on Computer Vision and Pattern Recognition. 2022.
> [4] Liu, Zechun, et al. "Reactnet: Towards precise binary neural network with generalized activation functions." Computer Vision–ECCV 2020: 16th European Conference, Glasgow, UK, August 23–28, 2020, Proceedings, Part XIV 16. Springer International Publishing, 2020.

---

> > ### Comment · Reviewer_FZQj · 2023-11-02
> > **Loss Landscape Visualization**
> >
> > The text at the top of page 5 appears too complicated, I suggest make it simple.

---

> > > ### Author Response · Authors · 2023-11-02
> > > **Response to Reviewer FZQj`s Comments on Paper1603 (2)**
> > >
> > > ### 9. Reorganization of the loss landscape visualization explanation text.
> > > Dear Reviewer FZQj, thanks again for your kind suggestion on simplifying the loss landscape visualization introduction part. Considering that we already have a table for the visualization algorithm, we have simplified text of this part and only kept a highly abstracted explanation for this process in the revised version.  Your kind suggestion would help readers gain a more comprehensive understanding of the main idea behind our work.

---

> ### Comment · Reviewer_FZQj · 2023-11-14
> **official recommendation for the submission**
>
> I recommend this paper for publication, contingent on the authors addressing the concern raised by Reviewer 3WQd.

---

> > ### Author Response · Authors · 2023-11-16
> > **Response to Reviewer FZQj`s Comments on Paper1603 (3)**
> >
> > we sincerely appreciate the reviewer's recommendation for this paper. We have revised the paper thoroughly and added all the related discussions during the revision to further improve the paper's quality.

---

### Review · Reviewer_3WQd · 2023-10-13

**Summary Of Contributions:**

The authors present a new approacht to BNN design, by including an element-wise multiplication after the first binary convolution block in each basic block (processing unit) and by feeding both the input and output of a Sign-Bconv-Bn-Prelu to the squeeze-and-expand unit. With that, the authors present a new (and larger) BNN network designs that allows for better optimization due to a smoother loss landscape. Next to model design, the authors find that teacher selection is crucial in applying knowledge distillation for training BNNs, and argue that progressively using stronger teachers during training aids the optimization process. Overall, some interesting finding are reported in the paper.

**Audience:**

Yes

**Claims And Evidence:**

Yes

**Requested Changes:**

[R1] The writing quality of the paper is a MAJOR concern, many typo's. Few examples:
- Page 2: "Infor-RCP" instead of "Info-RCP". "we show" with no capital.
- Page 6: "Fig. ?? and ??" are not referenced, space missing between "designLiu et al.", paper inconsistently uses "Sec. 6" vs "Section x"
- Page 7: "Accuray", "Specifically, We" -> no capital
- Table 3 is nowhere referenced in text, hence I have difficulties understanding what its purpose is.

[R2] Providing some sort of proof on why certain elements work is difficult in this context (understandably!), but I miss an intuitive explanation on why and how the authors think Info-RCP and ELM-attention help in smoothing the loss landscape. Specifically, it shows that the combination of the two items is key in smoothing the landscape. Both individually seem to marginally improve the smoothness of the landscape. To me, this is the main point of improvement - more experiments to elaborate on this point are required.

[R3] I understand from Table 6 that Consecutive KD virtually has no effect (+0.05), while there is extra training complexity associated with using more teachers. The text, however, does not reflect on the marginal impact of this improvement. Perhaps that the Consecutive KD has more impact for larger architectures (?), even that is unclear. Further elaboration on this matter are required.

[R4] I suggest adding a table to the ablation study which shows the added complexity of the different parts of the E2E architecture, with the associated accuracy gains.

[R5] The ablation study only discusses using or not using the ELM-attention. It would be interesting to investigate if it makes a difference WHERE the attention module is used. Right now only the first convolution unit in the basic blocks has the attention, would it benefit to apply it to the other units as well?

[R6] The paper currently does not report on-device latency as also mentioned in section 7.8. Implementing the architecture in a BNN framework like https://proceedings.mlsys.org/paper_files/paper/2021/hash/a0a81eed87dd44d6504fed5f81f6de5a-Abstract.html could help materialize this.

**Strengths And Weaknesses:**

Strength:
- The finding that teacher selection is an important step for knowledge distillation in BNNs and that too large knowledge gaps between student and teacher hinder the optimization of BNNs.

- The paper reaches SOTA and moves the Pareto-front for accuracy - model size trade-off.

- The paper addresses one of the main bottlenecks in BNNs: the rough loss landscape and the inherent difficulty of optimization.

- The addition of images of the first binary featuremaps is an interesting aspect. It helps corroborating the intuition that the network is indeed powerful enough to distinguish shadows and different shades of a particular color.

---

> ### Author Response · Authors · 2023-10-31
> **Response to Reviewer 3WQd`s Comments on Paper1603 (1)**
>
> **First and foremost**, we thank the reviewer for taking the time to provide thorough feedback on our manuscript. We appreciate the recognition of the strengths of our work and the detailed suggestions you provided. Below is our response to the specific points raised:
> ### **Responses to Weaknesses:**
>
> ### 1. The writing quality of the paper is a MAJOR concern, with many typos.
> **Response**: Thank you for pointing out the typographical errors and inconsistencies in the paper. We have undertaken a comprehensive review to ensure that all of these issues are corrected. *Table 3* is a summarization and comparison of the current BNN Design and optimization principle, and we have reorganized this part for a clearer understanding of its purpose.
>
> ### 2. Intuitive explanation of why and how the authors think Info-RCP and ELM-attention help in smoothing the loss landscape.
> **Response**: We acknowledge the importance of providing a more intuitive explanation behind the synergistic effect of Info-RCP and ELM-attention in smoothing the loss landscape. As we have analyzed in *section 3.1*, the discontinuity of the BNN loss landscape is largely influenced by the reduced tensor (weight, activation) representation and serious representation shift after each sign function. Info-RCP and ELM-attention work as a mixed channel-wise and spatial-wise attention together in the basic block design. The coupled attention design largely improves the representational capacity in forward information flow and enables each basic block to recalibrate the shifted features from two orthogonal dimensions, which is not possible to achieve with only one of them. Therefore, the current basic block implicitly rebuilds the continuous processing ability of Conv2d.  We hope this will provide a more institute understanding of the smoothness effect from both designs.
>
> ### 3. Elucidation on the Impact and Rationale Behind Consecutive KD.
> **Response**: We are grateful for the reviewer's astute observation on the marginal impact of Consecutive Knowledge Distillation (KD) as presented in *Table 6*, and their quest for a deeper understanding of this phenomenon. The essence of performing the ablation study encapsulated in *Table 6* was to unfold the intricacies of different training strategies, using BNext-T as a paradigm.
> In our Consecutive KD setup, Efficient-B2 was designated as the strong teacher, while Efficient-B0 was the assistant teacher. The data furnished in *Table 9* and *Figure 5* delineates the counter-intuitive overfitting predicament encountered during BNext model optimization, which is substantially tied to the knowledge complexity vested in teacher models. The relatively diminutive knowledge complexity gap (KC: 0.893 vs 1.103, single teacher KD accuracy for BNext-T: 71.32 vs 71.25) elucidates the minuscule enhancements rendered by additional Consecutive KD for BNext-T under identical hyperparameter configurations.
> However, it's imperative to spotlight that the Consecutive KD, despite its marginal numerical impact in *Table 6*, has underscored a pivotal revelation on mitigating the counter-intuitive overfitting issue as portrayed in *Table 9* and *Figure 5*. The reviewer's insight about the potential greater impact of Consecutive KD for larger architectures is indeed perspicacious and resonates with the findings presented in *Table 9*. The employment of Consecutive KD unveils a viable pathway to facilitate the distillation of larger binary architectures, thereby rendering modern teacher models like ConvNext viable for this endeavor.
> Through this discourse, we aim to provide a more nuanced understanding of the Consecutive KD's role and its nuanced impact under different architectural and training setups.
>
> ### 4. Adding a table to the ablation study that shows the added complexity of the different parts of the E2E architecture.
> **Response**: We appreciate the good suggestion for adding a table detailing the added complexity of different parts of the E2E architecture, paired with the accuracy gains. We have reorganized the *Table. 5* and show the extra complexity of adding ELM-Attention and Info-RCP modules on the BNext-Tiny backbone. Both designs are highly lightweight and contribute limited OPs and parameters. Taking ELM-Attention as an example, it is a parameter-free operation and only adds negligible element-wise multiplication within each basic block.

---

> > ### Author Response · Authors · 2023-10-31
> > **Response to Reviewer 3WQd`s Comments on Paper1603 (2)**
> >
> > ### 5. Ablation study on the placement of the ELM-attention module.
> > **Response**: Your suggestion on investigating the placement of the ELM-attention module is insightful. We have extended the ablation study to explore the impact of situating the attention module at different units within the architecture (BNext-18, BNext-Tiny). This will add depth to our analysis and offer a more comprehensive understanding of the module's role. Basically, our existing design shows the best performance and a potential explanation behind this would be a transformer-style structure in BNext basic block. However, we admit that a deeper exploration and understanding of the attention mechanism will further benefit BNN developments.
> >
> > ### 6. On-device latency for BNext is not reported as mentioned in the limitation.
> > **Response**: On-device latency is indeed a crucial metric for real-world applications. We recognize the benefits of reporting this and are working on implementing our architecture in the BNN framework you mentioned. The results from this will be included in the upcoming version.
> >
> > **In conclusion**, we thank the reviewer for the constructive feedback. We have revised our paper to solve the mentioned weakness and are confident that addressing these concerns has significantly improved the quality and clarity of our paper.

---

> > > ### Comment · Reviewer_3WQd · 2023-11-01
> > > **Curious to see on device performance report!**
> > >
> > > - Which table in the ablation study are you referring to? (P 5.)
> > > - I'm quite keen to see on device performance results. When do you expect you can report those? Can you please share here a version of that when ready? (P6.)

---

> > > > ### Author Response · Authors · 2023-11-02
> > > > **Response to Reviewer 3WQd`s Comments on Paper1603 (4)**
> > > >
> > > > ### 7. Adding the explanations in addressing point 2 here to the text.
> > > > We are happy that the feedback solve the main concern and appreciate reviewer`s suggestion on adding the explanation to the text. We have revised our paper and add some of the main claiments in point 2/3 to the paper for a better explanation of our core idea.
> > > >
> > > > ### 8. Rethinking Table 5 in terms of organization.
> > > > We agree with reviewer that the Table 5 may not be clear enough for readers and have improved it in the revised version. Thanks reviewer again for your kind help on the quality improvements of this paper.
> > > >
> > > > ### 9. Which table in the ablation study are you referring to? (P 5.)
> > > > Dear reviewer 3WQd, the new ablation study on the placement of element-wise attention is reported in Table 6, where we combine ELM-Attention design with different convolution module within each basic block of BNext-18 (ResNet-18 backbone) and BNext-T (MobileNet backbone variant proposed in ReActNet [1]).
> > > >
> > > > ### 10. On-device performance verification progress.
> > > > Dear reviewer 3WQd,  we are trying to report the ARM and CPU device`s performance with your suggested Larq library. However, Larq is based on TFLight and only support Tensorflow models, but BNext families were optimized using Pytorch, which causes some difficulty for the representation alignments and deployment. Right now, we are working on splitting the lower-level C++ code in Larq so that we can reimplement the BNext structure based on it.  This could take some time but we are confident that we can at least get a speed test in the coming days. The final results will also be added to the paper, considering that we still have some spaces on the main pages (12 page).
> > > >
> > > > [1] Liu, Zechun, et al. "Reactnet: Towards precise binary neural network with generalized activation functions." Computer Vision–ECCV 2020: 16th European Conference, Glasgow, UK, August 23–28, 2020, Proceedings, Part XIV 16. Springer International Publishing, 2020.

---

> > ### Comment · Reviewer_3WQd · 2023-11-01
> > **Good explanations, we need more actions!**
> >
> > - I suggest adding the explanations in addressing point 2 here to the text to help clarifying this further for the reader as well.
> > - I feel Table 5 is a bit cryptic with all those slashes (/) in between, I suggest rethinking this Table in terms of organization.
> > - Overall, good responses, and I hope we can reflect these into the text as much space allows.

---

> ### Comment · Reviewer_3WQd · 2023-11-04
> **Please apply those items in the final version. Thanks for your response.**
>
> With the condition that the above items are addressed and reflected in the final version, the paper is good to be considered for publication, in my view.

---

> > ### Author Response · Authors · 2023-11-16
> > **Response to Reviewer 3WQd`s Comments on Paper1603 (5)**
> >
> > ### 11 Apply those items in the Final version.
> > We would like to thank reviewer 3WQd again for the great guidance during the review process. We have revised our paper again to add more information about the on-device evaluation. All the interesting items from the above discussion are also collected and reflected in the final version. We believe addressing these concerns has significantly improved the quality and clarity of our paper.
> >
> > For on-device evaluation, we have assessed the real-world inference efficiency on a Banana Pi M5 device, utilizing the Larq Library as suggested by the reviewer. Notably, Larq predominantly supports TensorFlow and is limited to convolutions with regular channel multiples of 32. Despite these constraints, we successfully integrated Larq with PyTorch. This integration was achieved through the C++ extension API of PyTorch's official implementation, allowing for the compression of binary convolutions within BNNs. Other components of the neural network remain unaltered in this setup.
> >
> > In practice, the Larq Library achieves a 32x compression of weights in binary convolutions. This compression is executed by storing every 32 elements in an INT32 tensor using bit-packing, which is then unpacked during forward propagation. Binary matrix multiplication within these convolutions is optimized through the use of XNOR operations and bit-counting (ARM instruction sets), facilitating substantial acceleration.
> >
> > For instance, in our tests with a 1-bit convolution layer having an NCHW configuration of (16, 64, 128, 128), we observed a remarkable 20.1x speedup on the Banana Pi M5 with the same input shape, when compared to a native PyTorch CPU implementation. This evaluation was further extended by integrating the BNext-18 model, adhering to the channel constraints imposed by Larq. The input dimensions were set to [1, 3, 224, 224].
> >
> > Comparative analysis of inference times across different settings—pure PyTorch,  PyTorch+Larq C++, and PyTorch + Larq ARM instruction sets—yielded average times of 1.27s, 0.8948s, and 0.5759s, respectively. With the integration of the Larq Library, we achieved a maximum acceleration of 2.2 times, which is way less than a single binary convolution acceleration.
> >
> > However, it is important to note, as discussed in our Limitations section, that the current state of BNN acceleration lacks efficient implementation support from the community. The reliance on PyTorch's C++ extension API introduces constraints due to additional floating-point operations, which may not fully demonstrate the potential acceleration capabilities of BNNs. Though extremely optimizing for acceleration is out of the scope of this paper, we hope the good performance from BNext will motivate more research for the binary neural network community.

---

### Review · Reviewer_P96B · 2023-10-23

**Summary Of Contributions:**

The paper introduces a novel binary neural networks (BNNs) architecture termed BNext, accompanied by training methodologies including consecutive knowledge distillation (CKD). A comparison of loss landscapes across different BNN architectures is presented, with emphasis on the assertion that Info-RCP results in a smoother loss landscape. The paper includes thorough ablation studies, and the findings suggest that BNext achieves a commendable top-1 accuracy on the ImageNet dataset. A comparative analysis between the binary features of BNext-T and ReActNet is conducted, indicating that BNext-T possesses more varied binary features devoid of checkerboard artifacts.

**Audience:**

Yes

**Claims And Evidence:**

Yes

**Requested Changes:**

Please see weakness above.

**Strengths And Weaknesses:**

[Strength]
- The paper clearly articulates the observations and rationale behind the BNext architecture. The examination of design factors in BNNs that influence the loss landscape's smoothness is particularly enlightening. Figure 2 provides a helpful visual representation of the loss landscape, underscoring the potential advantages of BNext.
- The research offers comprehensive comparisons with numerous existing BNNs.
- Figure 6 distinctly demonstrates that BNext-T has the capability to produce superior binary features in contrast to ReActNet.
Given the same OPs or number of parameters, BNext models consistently deliver enhanced top-1 accuracy.

[Weakness]
- The reliance on CIFAR-10 and ImageNet datasets may seem a bit dated.
- The methodology primarily focuses on supervised learning. It prompts curiosity on whether the proposed concepts could be applied to visual Transformers, which are prevalent in contemporary applications.
- The utility of knowledge distillation seems to be more suitable for compact models operating under supervised learning guidelines.
- The authors should consider providing a more in-depth discussion on the method's limitations.

---

> ### Author Response · Authors · 2023-10-31
> **Response to Reviewer P96B's Comments on Paper1603**
>
> **First and foremost**, we sincerely appreciate the reviewer's detailed analysis of our paper. We understand the strengths and weaknesses pointed out and would like to address the concerns raised.
>
> ### **Strength Acknowledgment:**
> We are pleased to note that the reviewer has found the articulation of our observations and rationale behind the BNext architecture to be clear. We believe that understanding the influence of design factors on the smoothness of loss landscapes in BNNs is vital, and we're glad that our visual representations effectively emphasized the advantages of BNext. Moreover, we recognize the reviewer's appreciation of the comprehensive comparison made between BNext and existing BNNs.
>
> ### **Responses to Weaknesses:**
>
> ### 1. Choice of Datasets:
> **Response:** We acknowledge the reviewer's point regarding our reliance on CIFAR-10 and ImageNet datasets. These datasets have been pivotal in the domain of deep learning and the most recent BNN research due to their standard benchmarks on supervised classification tasks. We chose them to ensure consistency in comparison with the most recent BNN works [1,2,3]. However, we agree that there's room for incorporating more contemporary datasets. We will be happy if the reviewer can share some recommendations for this aspect. In future work, we aim to explore the performance of BNext on other datasets to showcase its adaptability and robustness.
>
> ### 2. Focus on Supervised Learning and Applicability to Visual Transformers:
> **Response:** Our choice to emphasize supervised learning was mainly due to its dominance in the current BNN research landscape. Nonetheless, the reviewer's curiosity regarding the applicability of our proposed concepts to visual Transformers is both valid and appreciated. The main observation during the design and optimization of BNext has shown the importance of smooth optimization, which can be achieved from the aspects of model design and improved supervision pipeline. Owing to the comprehensive consideration, BNext has already shown better performance than existing binary ViT work BiViT [2]. We concur that the methodologies embodied in BNext hold the potential to significantly bolster binary visual Transformers. This potential will be explored in our subsequent research endeavors. We do acknowledge an architectural divergence between CNNs and ViT. For instance, while ViT employs an unbiased transformer block to capture the data's inductive bias, the added binarization step, coupled with the ensuing gradient mismatch issue, may exacerbate the optimization challenges when compared to more conventional CNN architectures. Our research endeavors have shed light on this inherent complexity, offering initial guidelines for the more effective design and optimization of binary visual Transformers. Through these findings, we aspire to bridge the existent knowledge gap, propelling the discourse forward in this burgeoning field.
>
> ### 3. Utility of Knowledge Distillation for Compact Models:
> **Response:** We understand the reviewer's observation that the utility of knowledge distillation appears more tailored for compact models under supervised learning. However, our introduction of consecutive knowledge distillation (CKD) was to emphasize its advantages in binary neural networks, beyond just compact models. The CKD in BNext provides a way to harness deeper insights from teacher networks over consecutive iterations, especially in the case of strong teacher distillation. We'll aim to elucidate this further in our revised manuscript for clarity.
>
> ### 4. In-depth Discussion on Method's Limitations:
> **Response:** We agree with the reviewer's suggestion that a comprehensive discussion of the method's limitations is essential for a holistic understanding. Though we have considered this point in the appendix, we have delved deeper into the potential limitations, challenges, and scope of our methodology in the revised version. This will not only enrich the paper but also pave the way for future advancements.
>
> **In conclusion**, we express our gratitude for the thoughtful review. It has provided valuable insights that help improve the quality and clarity of our work. We are committed to addressing the raised concerns in the revised manuscript to ensure that our research is both comprehensive and future-oriented.
>
> **References:**
>
> [1] Zhang, Yichi, Zhiru Zhang, and Lukasz Lew. "Pokebnn: A binary pursuit of lightweight accuracy." Proceedings of the IEEE/CVF Conference on Computer Vision and Pattern Recognition. 2022.
>
> [2] He, Yefei, et al. "BiViT: Extremely Compressed Binary Vision Transformers." Proceedings of the IEEE/CVF International Conference on Computer Vision. 2023.
>
> [3] Liu, Zechun, et al. "Reactnet: Towards precise binary neural network with generalized activation functions." Computer Vision–ECCV 2020: 16th European Conference, Glasgow, UK, August 23–28, 2020, Proceedings, Part XIV 16. Springer International Publishing, 2020.

---

> > ### Comment · Reviewer_P96B · 2023-11-22
> > **Response from the reviewer**
> >
> > The authors have effectively addressed my concerns. I recommend this manuscript for publication, taking into consideration their responses to the other reviewers as well.

---

> > > ### Author Response · Authors · 2023-11-22
> > > **Response to Reviewer P96B's Comments on Paper1603 (2)**
> > >
> > > we sincerely appreciate reviewer P96B's great help and the recommendation for this paper.

---

### Decision · Action_Editor_6dHT · 2023-11-28

**Recommendation:** Accept with minor revision

**Comment:**

There were active discussions between the authors and Reviewer 3WQd. During the response period, the authors addressed all the reviewers' concerns and suggestions.

The revised paper looks okay to me, but I think this paper still should fit the standard of the official formatting instructions.

Now, all the citations seem to be declared by `\cite{}`. It should be `\citep{}` for better readability. For example, not "ResNet-18 He et al. (2016)", but " ResNet-18 (He et al. 2016)". Please check "4. Citations, figures, tables, references > 4.1 Citations within the text" of the official formatting instructions.

> When the authors or the publication are included in the sentence, the citation should not be in parenthesis, using \citet{} (as in “See Hinton et al. (2006) for more information.”). Otherwise, the citation should be in parenthesis using \citep{} (as in “Deep learning shows promise to make progress towards AI (Bengio & LeCun, 2007).”).

My recommendation is "Accept", but as this paper needs to be updated to reach the official formatting instructions, I recommend minor revision. I will check whether the final paper fits the standard.

**Audience:**

Lightweight and efficient neural networks are essential for many applications. Despite their importance, their performances are somewhat limited due to the difficulty of the optimization. This paper pushes the limitation of BNNs. In my opinion, this paper provides many interesting findings, and some audiences might be interested in the empirical findings of this work.

**Claims And Evidence:**

This paper aims to improve the training of BNNs. Two contributions are introduced: (1) a new architecture, BNext, (2) careful teacher selection for knowledge distillation (KD) for a new multi-round KD, named consecutive KD.

For supporting the first contribution, this paper proposes a new analysis based on the loss landscape visualization of various BNNs. From the observation, three core architectural design factors are introduced. For the second contribution, this paper introduces a new metric, named Knowledge Complexity (KC). From the analysis, this paper proposes to choose a teacher with a lower KC, rather than a high accuracy. In my opinion, each component is well-supported by proper analysis and ablation study in the revised paper.

The effectiveness of the proposed method is shown by the first BNN with over 80% ImageNet-1k accuracy. Also, following the comment from Reviewer 3WQd, this paper provides "On-Hardware Evaluation", that supports a more practical value of this work.

---

> ### Author Response · Authors · 2023-12-19
> **Camera-ready revision updated**
>
> Dear Action Editor,
>
> We would like to thank you for handling our paper, conveying the positive decision, and for the additional revision suggestion!
>
> we have meticulously revised the citations to adhere precisely to the official formatting instructions, as indicated by the use of the `\citep{}` command.
>
> Once again, thank you for your invaluable support throughout the review process.
>
> Best,
> Authors